# Model reference adaptive system and pseudo-sliding mode control with exponential reaching law for sensorless-speed control of PMSM

Djaloul Karboua[1]*, Toufik Mebkhouta[2], Said Benkaihoul[3], Youcef Chouiha[3], Belgacem Toual[1], Zuhair A. Alqarni[4], Ahmad F. Tazay[4], Mohamed I. Mosaad[5]

1 Renewable Energy Systems Applications Laboratory (LASER), University of Djelfa, Algeria,
2 Laboratory of Electrical Engineering Biskra (LGEB), Department of Electrical Engineering, Faculty of Science and Technology, University of Biskra, Algeria, 3 Laboratory of Applied Automation and Industrial Diagnostics (LAADI), Department of Electrical Engineering, Faculty of Science and Technology, University of Djelfa, Algeria, 4 Electrical Engineering Department, Al-Baha University, Al Baha, Saudi Arabia, 5 Electrical and Electronic Engineering Department, Yanbu Industrial College, Yanbu, Saudi Arabia

* djaloul.karboua@univ-djelfa.dz

## Abstract

Sensorless speed motor drives are essential for developing control techniques, reducing cost, streamlining the system, and enhancing reliability. This study explores the area of sensorless speed control for Permanent Magnet Synchronous Motors (PMSMs) by proposing a hybrid control technique. This technique integrates the model reference adaptive system (MRAS) and pseudo-sliding mode control with an Approach Reaching Law (ARL). The MRAS is implemented as a robust sensorless observation method that efficiently handles uncertainties, adapts to dynamic conditions, and aims to achieve dependable performance by having the controlled system mimic a reference model. On the other hand, the pseudo-sliding mode control entails using a continuous approximation (CA) methodology to successfully resolve the chattering problem often seen in conventional sliding mode control approaches. This approach of control allocation enables more seamless control signals, hence improving the longevity of the system and minimizing unwanted oscillations. The ARL component relies on the Exponential Reaching Law (ERL), which enables fast and precise convergence to the intended sliding surface. The exponential characteristics of the ERL lead to faster reaction times and enhanced dynamic performance, guaranteeing the timely attainment of the sliding surface while being robust against changes in parameters and external disturbances. To assess the efficacy of the proposed sensorless hybrid control approach, uncertainties and disturbances were simulated, including PMSM parameters variation, load torque application, and speed level changes. The effectiveness of the suggested approach is compared to control strategies for PMSM, such as the classical ERL-SMC (Type 1) and the pseudo-sliding mode ERL-SMC (Type 2) schemes. The simulations were conducted exclusively using MATLAB/Simulink.

**Data availability statement:** All relevant data are within the manuscript.

**Funding:** The author(s) received no specific funding for this work.

**Competing interests:** The authors have declared that no competing interests exist.

## 1. Introduction

The demand for high-performance and efficient motor control systems has intensified with the increasing integration of electric motors in various applications, ranging from industrial machinery to electric vehicles. Among the various motor types, Permanent Magnet Synchronous Motors (PMSMs) have garnered significant attention due to their favorable characteristics, including high efficiency, power density, and precise control capabilities. However, PMSM requires precise control and observation to ensure optimal performance, efficiency, and stability across various operating conditions. Accurate speed and torque control is essential for applications demanding high precision and dynamic response, such as robotics, electric vehicles, and industrial automation. Observation techniques, particularly in sensorless configurations, play a critical role by estimating the rotor position and speed from electrical signals, thus eliminating the need for physical sensors. The primary goals of sensorless speed control include reducing system cost and complexity, improving reliability, and enhancing the motor's ability to operate in harsh environments without sensor degradation. The current techniques for sensorless PMSM control can be categorized into two groups based on their operational speed range. The first group includes techniques that operate in the zero to low-speed range of 0–10% of base speed, while the second group includes techniques that operate in the medium speed range of 10–100% of base speed, [1–5] and high-speed range (100–200% of base speed) [6].

The primary categories and classifications for the sensorless PMSM control are depicted in Fig 1. In the context of sensorless control, machine model-based techniques are generally used more frequently in the medium and high-speed range compared to non-model-based techniques [7]. Notably, non-model-based strategies include those based on artificial intelligence [8,9]. Alternatively, there are machine model-based techniques that can be either adaptive or non-adaptive [1,10]. The inability to operate at speeds ranging from zero to low is a common characteristic shared by these many techniques [11]. Saliency-based approaches are sometimes called High Frequency Model (HFM) methods. Within the range of zero to low speed, they are employed to provide sensorless control [5]. The machine working theory is based on the HFM [5]. To estimate the position of the rotor, an HF signal is fed, using the anisotropy of the magnetic circuit caused by the geometric or magnetic saliency of the rotor [12,13]. These techniques are advantageous because they are efficient while the machine is operating at low speeds or when it is completely stopped. This advantage does not depend on the rotor speed [14]. However, the drawback of this is that the injection of HF signals into the machine leads to torque ripples [15]. According to these studies, employing PMSMs in low-speed range poses some challenges, notwithstanding its intrinsic benefits. A primary difficulty is cogging torque, resulting from the contact between the rotor magnets and the stator slots. This may result in irregular motion at low velocities, causing vibrations and diminished efficacy in precise applications. Thermal management is an additional problem. Operating at low speeds diminishes the efficacy of cooling systems since several processes depend on elevated rates for sufficient airflow. This may result in overheating, especially in

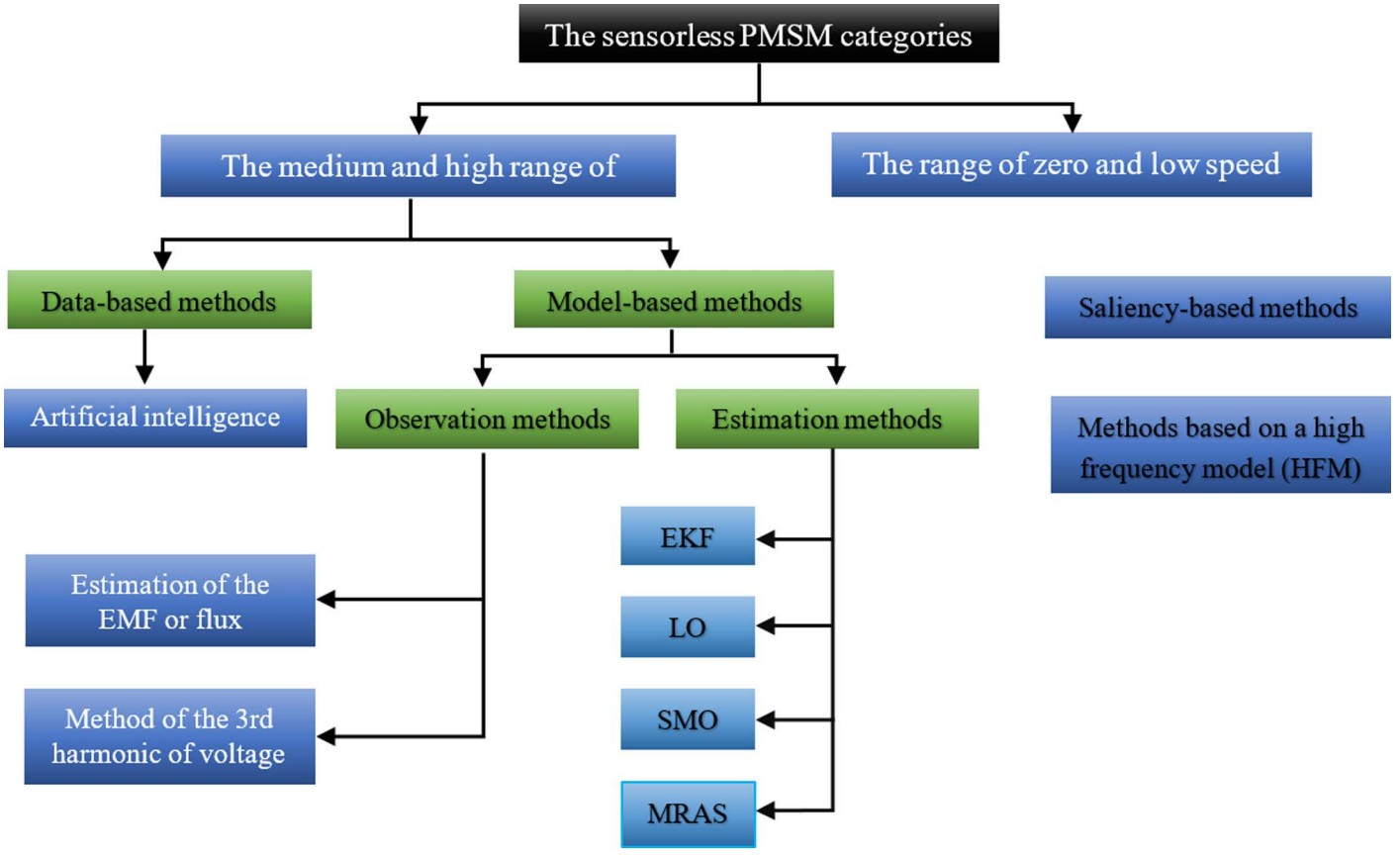

**Fig 1. Major categories and classifications of sensorless PMSM.**

high-torque situations. Moreover, current harmonics resulting from suboptimal inverter switching or control errors may impair motor performance and efficiency at reduced velocities.

Data-based methods are alternatively referred to as based methods. These are employed in the medium and high-speed ranges for sensorless control in artificial intelligence-based systems. Fuzzy logic, neural networks, and machine learning form the fundamental basis of their operational principles. These techniques have the advantage of not necessitating an analytical model of the machine. However, its downsides include a somewhat long learning curve and complexity in implementation [5,16].

Model-based methods, designed for sensorless control in the medium and high-speed range, principally rely on estimating some values (such as magnetic flux or EMF) that are dependent on the position of the rotor [17]. To achieve this, the voltages and currents of the machine are measured [1]. These strategies have the benefits of being user-friendly and using minimal computational resources. However, their ability to withstand and accurately measure may be reduced due to their vulnerability to interference from measuring probes. There are two types of approaches when it comes to machine-model-based techniques: non-adaptive and adaptive. Ultimately, we aim to distinguish between the processes of estimation and observation [18].

- **Estimation methods (EM):** This category is also called no-adaptive. Their method is based on directly estimating the machine's quantities from its state model, which includes the rotor information (EMF or flux). This is carried out without using a corrective term to make up for the discrepancy between the machine-measured quantities and the values

predicted by the model. Thus, they realize that they are highly susceptible to the machine's parametric uncertainties. Non-adaptive techniques are therefore categorized as "Estimators"[1,19].

- **Observation methods (OM):** This category is also called adaptive. The sole differentiation between estimation methods lies in the utilization of a correction term. Their operational method involves utilizing a corrective term to align the disparities between the quantities measured by the machine and the quantities estimated by the state model. Indeed, this ensures that the calculated amounts will converge to the observed values. Therefore, these strategies demonstrate resilience in the face of ambiguity regarding the machine's parameters. Observation techniques are classified as "adaptive" because of the outcome they produce [1,19].

The robustness of the algorithms is a critical feature for situations where sensorless control is needed to guarantee operation continuation in both normal mode and degraded mode. This resilience pertains to changes in the load profile and machine parameters. The sensorless control of the integrated machine must ensure a superior degree of performance in terms of durability, precision, and ease of installation in this scenario. Upon reviewing the criteria and examining the advantages and disadvantages of the approaches indicated in Fig 1, it can be concluded that the observation methods are the most suitable.

The presented study aims to develop sensorless control algorithms for integrated machines that operate at medium and high speeds. In addition, state observers offer a trade-off between the accuracy of quantity estimation and the ease of implementing the algorithm (for rotor position and speed) for PMSM machines. It is widely recognized that utilizing saliency-based methods in machine learning is the only effective way to address the problem of observing PMSM machines operating at very low speeds in the zero to low speed range [20]–[21].

The observation method is considered crucial in applications where it is applied for sensorless speed, where adding physical speed sensors is impractical or cost-prohibitive. These techniques leverage the inherent characteristics of the motor and system dynamics to accurately speed estimation, hence enhancing the efficiency and cost-effectiveness of electric drives and motor control systems. Numerous studies have examined the use of state observers for sensorless control of PMSM machines. In the medium and high-speed range, their application to three-phase PMSM machines has shown positive results [6,17,22]. These techniques operate by doing a real-time comparison between the machine's actual measured current and the analytically predicted one.

In fact, minimizing the error between the estimated and observed currents is the primary goal of the algorithms that represent these observers. This is achieved by modifying the adjustable parameters of the observers in order to align the anticipated amounts more closely with the machine's measurements [6,19]. Realizing that an accurate assessment of the machine's state quantities accurately determines the rotor's position. Through the literature, the observation techniques that applied for the sensorless speed have been classified among: the Extended Kalman Filter (FKE), the Luenberger Observer (LO), the sliding mode observer (SMO), and the Model Reference Adaptive System (MRAS) [23–26]. The algorithms of these observers are constructed using either the frame (d-q) or the frame (α-β) of the machine model.

On the other hand, control techniques for PMSMs typically involve sophisticated methods designed to regulate current and speed accurately. A prominent approach is field-oriented control, which transforms the motor's three-phase currents and voltages into a two-coordinate system aligned with the rotor flux vector. This transformation allows for independent control of motor parameters, facilitating precise management of motor dynamics [27–30].

Another notable technique is Sliding Mode Control SMC, which ensures robust performance by driving the system towards a predefined sliding surface. SMC offers significant benefits due to its robustness in facing disturbances and uncertainties, making it particularly suitable for applications requiring high precision and stability. While classical SMC is known for its robustness, it encounters several challenges and weaknesses. One major issue is "chattering," stemming from the rapid switching of the control signal. This phenomenon can cause mechanical systems to experience excessive wear, unwanted vibrations, and instability due to high-frequency oscillations. Moreover, the discontinuous nature of

the control law poses difficulties in digital control systems, which are limited by sampling rates and actuator bandwidths, hindering the realization of optimal sliding mode behavior. Another significant concern is the susceptibility to measurement noise. The reactive nature of the switching control law amplifies any noise in system state measurements, exacerbating chattering. Additionally, designing the sliding surface and ensuring efficient reaching of this surface for system states can be intricate, especially for complex or nonlinear systems. This process necessitates meticulous tuning and a profound grasp of system dynamics, proving challenging and time-consuming. Despite these hurdles, various enhancements such as pseudo-sliding mode, higher-order sliding modes, and approach-reaching laws have been introduced to alleviate these issues, thereby improving the practicality and performance of SMC in real-world applications [31–37].

This paper primarily focuses on optimizing the medium to high-speed operation of PMSMs due to the critical demand for precision and dynamic performance in these speed ranges. While the challenges inherent to low-speed operation, such as accurate rotor position estimation, are recognized, it is noted that sensorless control systems encounter significant difficulties at low speeds due to reduced back-EMF signals. Techniques like saliency-based High Frequency (HF) injection methods have proven effective by leveraging the rotor's anisotropic properties for position estimation when conventional back-EMF sensing is ineffective. However, these methods often introduce drawbacks, such as torque ripple and noise, which can impact motor stability and performance.

To address these challenges, this work proposes a hybrid control strategy for the speed and current loops of PMSMs by combining pseudo-SMC with an Adaptive Reaching Law (ARL) and integrating a MRAS-based sensorless estimation. The pseudo-SMC component utilizes a CA technique to mitigate the chattering typically associated with traditional SMC, resulting in smoother control signals, improved system longevity, and minimized oscillations. The ARL, designed using the ERL, enables rapid and precise convergence to the sliding surface, enhancing dynamic response and ensuring robustness against parameter variations and external disturbances. Incorporating MRAS-based sensorless estimation further reduces system complexity, cost, and maintenance requirements by eliminating physical speed sensors. MRAS employs mathematical models and adaptive algorithms for accurate speed estimation, providing reliable speed control even in dynamic and uncertain environments.

The tuning of critical parameters, such as the gains of the PI regulator in MRAS and the sliding surface in SMC, is essential to achieving optimal system performance. This process involves iterative simulations and experimental validation to ensure stability, precision, and dynamic response across varying operational conditions. Furthermore, the computational complexity of implementing the proposed hybrid control strategy has been carefully analyzed, with optimizations designed to ensure its feasibility for real-time applications. These considerations balance robust control performance and practical implementation, making the system suitable for diverse PMSM applications.

Although the focus is on medium to high-speed operation, this strategy's adaptability and robust observation techniques indicate the potential for extending effective control to low-speed conditions, with future work to further minimize torque ripple and enhance system stability.

## 2. Constructing a model of the pmsm system

A PMSM functions like a three-phase induction motor. The three-phase voltage source connected to the stator windings generates a rotating magnetic flux. In a PMSM, the number of poles on the stator and rotor are equal since the rotor is a magnet, resulting in a constant rotor flux. This characteristic reduces brush and rotor losses by eliminating the need for the source to supply an excitation current [38]. The voltage applied to the machine can be described in the d-q reference frame using the motor torque, as shown in Eqs (1) and (2), respectively [39,40]:

$$\begin{cases} v_d = R_s.i_d + L_d\frac{di_d}{dt} + E_d \\ v_q = R_s.i_q + L_q\frac{di_q}{dt} + E_q \end{cases}$$

$$(1)$$

$$\begin{cases} j \cdot \frac{d\omega_m}{dt} = T_e - T_l - F\omega_m \\ T_e = (3 \cdot \frac{p}{2})((L_d - L_q)\, i_d \cdot i_q + \psi_f \cdot i_q) \\ \omega_e = p \cdot \omega_m \\ \frac{d\theta}{dt} = \omega_m \end{cases} \qquad (2)$$

Where, $i_d, i_q$ are d-q axis equivalent stator currents; $v_d, v_q$ are d-q axis equivalent stator voltages; $R_s$ is per phase stator resistance; $L_d, L_q$ are d-q axis equivalent stator and $E_d, E_q$ are d-q axis back-EMFs.

The $E_d$ and $E_q$ are presented as follows:

$$\begin{cases} E_d = -L_q \cdot \omega_e \cdot i_q \\ E_q = L_d \cdot \omega_e \cdot i_d + \omega_e \cdot \psi_f \end{cases} \qquad (3)$$

Where $\omega_e$ and $\omega_m$ are an electrical (stator) speed; $\psi_f$ is rotor magnetic flux linking the stator, $\theta$ is the angular speed of the rotor, $p$ is the number of pole pairs; $T_e$, $T_l$ are electromagnetic and load torques, respectively; j is the moment of inertia of the rotor; F is the friction constant of the rotor and $\theta$ is the angular speed of the rotor.

## 3. Designing of PMSM sensorless speed observation using MRAS

The MRAS-based sensorless speed observation system is intended to handle load situation fluctuations and uncertainty in the motor characteristics. Adaptive observation increases the robustness and dependability of the observation system by dynamically adapting to changes in the motor properties by utilizing feedback from the motor. To implement the MRAS, two independent models should be used. The first model is the reference model, which is used to calculate the stator currents' two components in the direct and quadrature axes ($i_d$ and $i_q$) concerning the Park frame of reference through direct measurements of reference stator currents. The second one is the adjustable model, which is used to estimate the two stator current components ($\hat{i}_d$ and $\hat{i}_q$) through direct measurements of stator currents and voltages ($v_d$ and $v_q$). We can estimate the rotor speed in the dynamic domain by cancelling the difference between the reference model's and the adjustable model's stator currents. The adaptive process creates the estimated value and causes it to converge towards the reference value using this difference. The behavior of the adaptive model tends to resemble that of the reference model due to an adaptation mechanism, usually a PI regulator. Fig 2 shows the structure of the PMSM sensorless speed observation using the MRAS method [41].

Selecting a reference frame that is connected to the rotor is essential in order to use the MRAS technique to estimate the stator resistance and the rotor speed. The rotor position, as determined by the reference model's adaptive technique, is utilized in this transformation. The Clark transformation (α-β), which employs the axis coordinates (d-q), is necessary to have sensorless control in a frame of reference connected to the stator, given that the rotor's initial location is precisely known [41].

### 3.1. MRAS observer equations

Using measurements of stator currents and voltages, we create two estimators of stator currents in this frame, based on the dynamic model of the synchronous machine with magnets, in a frame linked to the rotor (d- q). We obtain the state model of the PMSM represented in the reference frame connected to the rotor after transforming and organizing the equations by performing the required manipulations and transformations [6]:

$$\begin{cases} \frac{d}{dt}\begin{bmatrix} i_d & i_q \end{bmatrix}^T = \begin{bmatrix} -R_s/L_d & \omega_e \\ -\omega_e & -R_s/L_q \end{bmatrix} \cdot \begin{bmatrix} i_d & i_q \end{bmatrix}^T + \begin{bmatrix} 1/L_d & 0 \\ 0 & 1/L_q \end{bmatrix} \cdot \begin{bmatrix} v_d & v_q \end{bmatrix}^T + \begin{bmatrix} 0 \\ -\omega_e \frac{\psi_f}{L_q} \end{bmatrix} \\ \begin{bmatrix} y_1 & y_2 \end{bmatrix}^T = \begin{bmatrix} 1 & 0 \\ 0 & 1 \end{bmatrix} \cdot \begin{bmatrix} i_d & i_q \end{bmatrix}^T \end{cases}$$

$$(4)$$

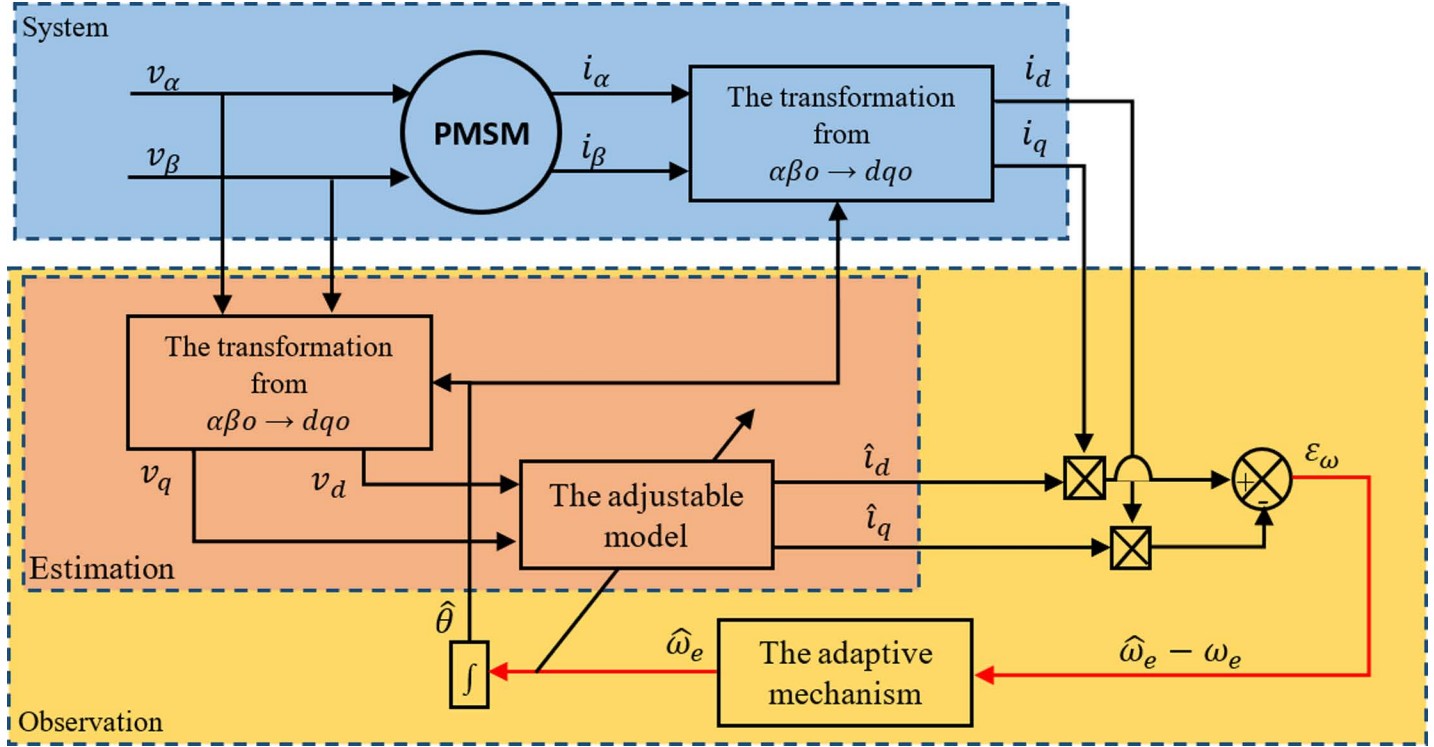

**Fig 2. PMSM sensorless speed observation principle by MRAS.**

We now construct two stator current estimators from the measurement of stator voltages and currents. The first is constructed from the system, (1) and the second from the system (4), such that:

$$\begin{cases} i_d = i_\alpha \cos \hat{\theta}_e + i_\beta \sin \hat{\theta}_e \\ i_q = -i_\alpha \sin \hat{\theta}_e + i_\beta \cos \hat{\theta}_e \end{cases}$$

(5)

The system used to determine the stator currents from the observed currents is kept as a reference model since it is independent of the rotor speed, $\omega_e$ [6]. We treat the system (Eq (1)) as an adjustable model as the estimators of the stator currents from the stator equations rely on the rotor speed, $\omega_e$ and the stator resistance, $R_s$.

$$\begin{cases} \frac{d}{dt} \begin{bmatrix} \hat{i}_d & \hat{i}_q \end{bmatrix}^T = \begin{bmatrix} -R_s/L_d & \hat{\omega}_e \\ -\hat{\omega}_e & -R_s/L_q \end{bmatrix} \cdot \begin{bmatrix} \hat{i}_d & \hat{i}_q \end{bmatrix}^T + \begin{bmatrix} 1/L_d & 0 \\ 0 & 1/L_q \end{bmatrix} \cdot \begin{bmatrix} v_d & v_q \end{bmatrix}^T + \begin{bmatrix} 0 \\ -\hat{\omega}_e \frac{\psi_f}{L_q} \end{bmatrix} \\ \begin{bmatrix} y_1 & y_2 \end{bmatrix}^T = \begin{bmatrix} 1 & 0 \\ 0 & 1 \end{bmatrix} \cdot \begin{bmatrix} \hat{i}_d & \hat{i}_q \end{bmatrix}^T \end{cases}$$

(6)

We then describe the differences in the stator currents in a reference linked to the rotor, as follows, knowing that the two models (reference and adjustable) use the same inputs (stator voltages):

$$\begin{cases} \varepsilon_d = i_d - \hat{i}_d \\ \varepsilon_q = i_q - \hat{i}_q \end{cases}$$

(7)

$$\begin{cases} \frac{d\varepsilon_d}{dx} = -\frac{R_s}{L_d}\varepsilon_d + \frac{L_q}{L_d}\omega_e.i_q - \frac{L_q}{L_d}\hat{\omega}_e.\hat{i}_q \\ \frac{d\varepsilon_q}{dx} = -\frac{R_s}{L_q}\varepsilon_q - \frac{L_d}{L_q}\omega_e.i_d + \frac{L_d}{L_q}\hat{\omega}_e.\hat{i}_d + \frac{\psi_f}{L_q}\hat{\omega}_e - \frac{\psi_f}{L_q}\omega_e \end{cases} \tag{8}$$

Upon finalizing the system (8) by introducing and removing the term $\frac{L_q}{L_d}\hat{\omega}_e.i_q$ and $\frac{L_d}{L_q}\hat{\omega}_e.i_d$ on the direct and quadratic axes respectively, we arrive at:

$$\begin{cases} \frac{d\varepsilon_d}{dx} = -\frac{R_s}{L_d}\varepsilon_d + \frac{L_q}{L_d}\hat{\omega}_e.\varepsilon_q + \frac{L_q}{L_d}i_q(\omega_e - \hat{\omega}_e) \\ \frac{d\varepsilon_q}{dx} = -\frac{R_s}{L_q}\varepsilon_q - \frac{L_d}{L_q}\hat{\omega}_e.\varepsilon_d - \frac{L_d}{L_q}i_d(\omega_e - \hat{\omega}_e) - \frac{\psi_f}{L_q}(\omega_e - \hat{\omega}_e) \end{cases} \tag{9}$$

In matrix writing, the deviations of the stator currents become:

$$\frac{d}{dt}\begin{bmatrix} \varepsilon_d \\ \varepsilon_q \end{bmatrix} = \begin{bmatrix} -\frac{R_s}{L_d} & \frac{L_q}{L_d}\hat{\omega}_e \\ -\frac{L_d}{L_q}\hat{\omega}_e & -\frac{R_s}{L_q} \end{bmatrix} \begin{bmatrix} \varepsilon_d \\ \varepsilon_q \end{bmatrix} + \begin{bmatrix} \frac{L_q}{L_d} \\ -\frac{L_d}{L_q}i_d - \frac{\psi_f}{L_q} \end{bmatrix} (\omega_e - \hat{\omega}_e) \tag{10}$$

Ultimately, the estimation error can be expressed as follows in state equation form:

$$s[\varepsilon] = [A][\varepsilon] + [W] \tag{11}$$

Where; $[\varepsilon]$ is the difference between the reference model and the adjustable model, $[W]$ is the feedback block, which constitutes the input to the linear block and s is a differential operator.

Fig 3 depicts a nonlinear counter-reaction system made up of Eqs (8) and (10). In fact, a nonlinear portion with input $\varepsilon(t)$ and output $W(\varepsilon, t)$ and a linear block specified by the transfer matrix $H(s) = (s[I] - [A])^{-1}$ can be used to schematize this system.

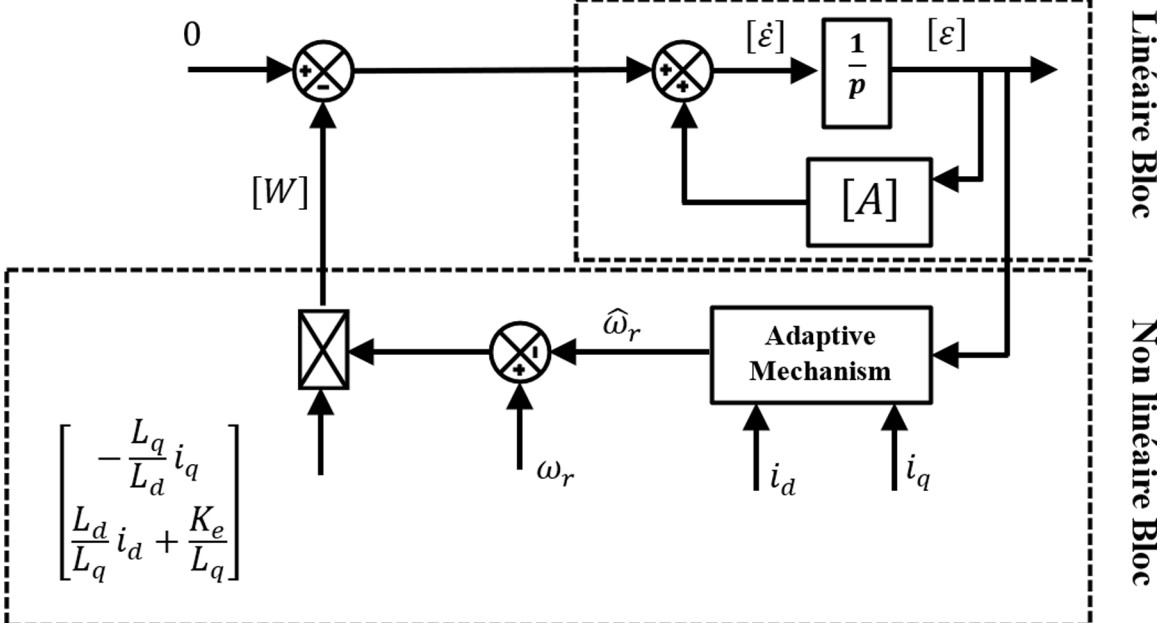

**Fig 3. MRAS in the form of a looped system.**

The counter-reaction system must meet two requirements to be considered hyper-stable: either $H(s)$ must be a strictly positive real matrix, or equivalently, all of the components' poles must have negative real portions.

## 3.2. Stability study of the MRAS observer

Our system's stability issue will be resolved by applying Popov's hyperstability notion, first presented in the early 1970s [42]. For the feedback block to be deemed hyper stable, it must meet Popov's inequality requirements:

$$\int_0^{t_1} [\varepsilon]^T [W] dt \geq -\chi^2 \text{ For } t_1 \geq 0 \tag{12}$$

where; $\chi$ is a positive constant.

The rotor speed estimate is given by:

$$\hat{\omega}_e = A_2([\varepsilon]) + \int_0^{t_0} A_1([\varepsilon]) dt \tag{13}$$

Where; $A_1$ and $A_2$ are nonlinear functions of the $\varepsilon_d$ and $\varepsilon_q$.

Using the expression for $[W]$, the quantity Eq (9) is equivalent to:

$$\int_0^{t_0} \left\{ \left[ \varepsilon_d \left( \frac{L_q}{L_d} i_q \right) \right] - \left[ \left[ \varepsilon_q \left( \frac{L_d}{L_q} i_d + \frac{\psi_f}{L_q} \right) \right] \right] \right\} [\omega_e - \hat{\omega}_e] dt \geq -\gamma_0^2 \tag{14}$$

Using Eq (10), Popov''s criterion for the current system becomes:

$$\int_0^{t_0} \left\{ \left[ \varepsilon_d \left( \frac{L_q}{L_d} i_q \right) \right] - \left[ \left[ \varepsilon_q \left( \frac{L_d}{L_q} i_d + \frac{K_e}{L_q} \right) \right] \right] \right\} \int_0^{t_0} \left[ \omega_e - A_2([\varepsilon]) - \int_0^{t_0} A_1([\varepsilon]) d \right] dt \geq -\gamma_0^2 \tag{15}$$

And applying the following inequality [39]:

$$\int_0^{t_0} k [pf(t)] f(t) dt \geq -\frac{1}{2} kf(0)^2 \qquad , k > 0 \tag{16}$$

Comparing Eq (13) and Eq (9), the expressions of $A_1$ and $A_2$ are expressed as follows:

$$\begin{cases} A_1 = K_1 \left[ \frac{L_q}{L_d} \varepsilon_d i_q - \frac{L_d}{L_q} \varepsilon_q i_d - \frac{\psi_f}{L_q} \varepsilon_q \right] \\ A_2 = K_2 \left[ \frac{L_q}{L_d} \varepsilon_d i_q - \frac{L_d}{L_q} \varepsilon_q i_d - \frac{K_e}{L_q} \varepsilon_q \right] \end{cases} \tag{17}$$

where; $K_1$ and $K_2$ are positive constants called adaptation gains.

The estimation of the rotation speed, $\hat{\omega}_e$ is a function of the error $\varepsilon$, as can be seen from the overall structure of the adaptation mechanism. We will estimate the rotor speed using a PI regulator to enhance the adaptation algorithm's reaction. Thus, the rotation speed estimate is provided in the format below [41]:

$$\hat{\omega}_e = K_{i\omega_e} \int_0^t \left( \frac{L_q}{L_d} \varepsilon_d i_q - \frac{L_d}{L_q} \varepsilon_q i_d - \frac{\psi_f}{L_q} \varepsilon_q. \right) dt + K_{p\omega_e} \cdot \left( \frac{L_q}{L_d} \varepsilon_d i_q - \frac{L_d}{L_q} \varepsilon_q i_d - \frac{\psi_f}{L_q} \varepsilon_q \right) + \hat{\omega}_e(0) \tag{18}$$

With $K_{i\omega_e}$ and $K_{p\omega_e}$ are the gains of the PI regulator to correct the error between the real speed and the estimated one.

Finally, the estimated electrical position of the rotor is obtained by integrating the estimated rotor speed.

$$\hat{\theta}_r = \int_0^t \hat{\omega}_e dt + \theta_{e0} \tag{19}$$

With $\theta_{e0}$ representing the initial condition on the estimated electrical position.

It is thus possible to determine the appropriate adaptation mechanism by applying Popov's hyperstability criterion [42]. The stability of the global system is ensured by this basic law. We assume in this phase that the two parameters, $\hat{\omega}_e$ and $\omega_e$, are time-varying, and each of them can be thought of as an input to the stator Eq (10)). The stator equations must be linearized for a modest fluctuation around an operational point in order to examine the dynamic response of the rotor speed estimation. As a result, the following equation describes how the inaccuracy varies:

$$\Delta\varepsilon(t) = \varepsilon(t) - \varepsilon(t - \Delta T) \tag{20}$$

where; $\Delta T$ represents the time interval over which the change in the variable $\varepsilon(t)$ is being measured.

The transfer function that links $\Delta\varepsilon_\omega$ and $\Delta\hat{\omega}_e$ will be calculated using this feature. The transfer function that links $\Delta\varepsilon_\omega$ and $\Delta\hat{\omega}_e$ can be obtained from Eqs (7 and 21):

$$\left.\frac{\Delta\varepsilon_\omega}{\Delta\hat{\omega}_e}\right|_{\Delta\omega_e=0} = G_{p\_\hat{\omega}_e} = \frac{K_L^2(s+T_d)+K_L i_q \omega_e}{[(s+T_d)(s+T_q)+\omega_e^2]} \tag{21}$$

Where;

$$K_L = \frac{\psi_f}{L_q}; T_d = \frac{R_s}{L_d} \ et \ T_q = \frac{R_s}{L_q}$$

Fig 4 provides the closed-loop block diagram for the dynamic response of the rotor speed estimation using the MRAS approach.

From Fig 4, we obtain the transfer function that connect $\omega_e$ and $\Delta\hat{\omega}_e$

$$\frac{\hat{\omega}_e}{\Delta\hat{\omega}_e} = G_{p\_\hat{\omega}_e}\left(\frac{K_{p\hat{\omega}_e}+K_{i\hat{\omega}_e}}{p}\right) \tag{22}$$

## 4. Design of PMSM control based on the ERL-SMC

SMC is a robust variable structure nonlinear control technique that switches between two values based on a specific switching logic S(x). This method is particularly effective for nonlinear systems across various fields, such as PMSM. SMC is highly resilient to parameter variations and excels in interference suppression and dynamic performance, ensuring a robust closed-loop system even under uncertainties and external disturbances [43–45]. The design of an SMC involves two key steps: driving the system states towards a predefined sliding surface and then maintaining the system dynamics

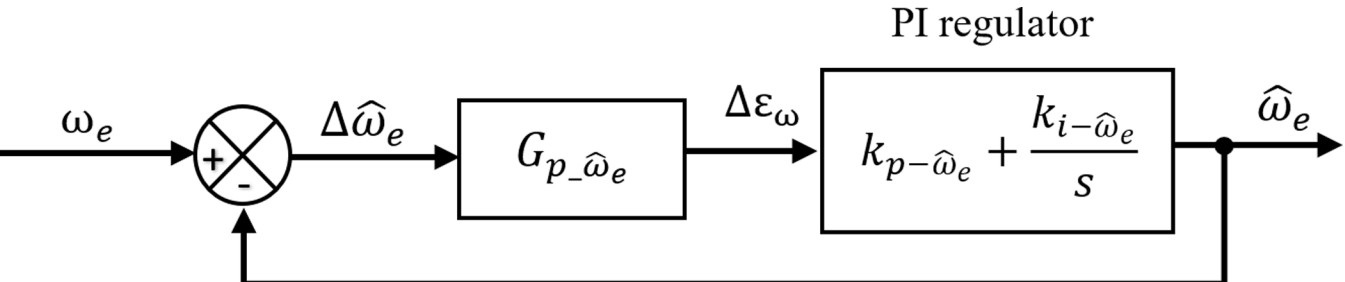

**Fig 4. Closed-loop block diagram of rotor speed estimation.**

along this surface to achieve equilibrium [46,47]. However, classical SMC, while powerful and robust, faces several challenges and weaknesses. One of the primary issues is the phenomenon known as "chattering," which is caused by the high-frequency switching of the control signal. Chattering can lead to excessive wear and tear in mechanical systems, generate unwanted vibrations, and induce high-frequency oscillations that can affect the system's performance and stability. Additionally, the discontinuous nature of the control law in classical SMC can be difficult to implement in digital control systems due to the limitations in sampling rates and actuator bandwidths. This can lead to practical difficulties in achieving the ideal sliding mode behavior. Another significant problem is the sensitivity to measurement noise. Since the switching control law reacts strongly to changes in the system's state, any noise in the state measurements can cause unnecessary switching, exacerbating the chattering problem. Furthermore, designing the sliding surface and ensuring the system states reach this surface efficiently can be complex, particularly for high-dimensional or highly nonlinear systems. The design process requires careful tuning and a deep understanding of the system dynamics, which can be challenging and time-consuming. Despite these challenges, various modifications and improvements, such as pseudo-sliding mode, higher-order sliding modes and approach reaching Law (ARL), have been developed to mitigate these issues and enhance the practicality and performance of SMC in real-world applications [31–37]. In this work, we apply a hybrid control strategy for the speed and current loops of a PMSM by combining pseudo-SMC with an ARL. The pseudo-sliding mode component employs a continuous approximation (CA) technique, which mitigates the chattering issue commonly associated with traditional sliding mode control. This continuous approximation ensures smoother control signals, enhancing system longevity and reducing unwanted oscillations. Meanwhile, the ARL component is based on the Exponential Reaching Law (ERL), which provides rapid and precise convergence to the desired sliding surface. The exponential nature of the ERL facilitates a faster response and improved dynamic performance, ensuring the system reaches the sliding surface quickly and maintains robustness against parameter variations and external disturbances. This hybrid approach leverages the benefits of both techniques, offering a robust, stable, and efficient control solution for PMSM applications. Generally, implementing this kind of control requires three main actions.

### 4.1. Choice of the sliding surface

Establish a suitable sliding surface in the state space that reflects the desired system behavior. This surface acts as the target trajectory for the system states. According to J.J. Slotine's proposition, the general formula for designing sliding surfaces is as follows [48–51]:

$$S(x,t) = \left(\frac{d}{dt} + \lambda\right)^{n-1} e(t) \tag{23}$$

Where $\lambda$ is a positive number chosen by the designer (scaling factor), $S(x,t)$ is the sliding surface, $e(t)$ is the tracking error between the reference and actual state variable, and n is the system's order.

The tracking error expression to the PMSM is obtained as follows:

$$e(x,t) = x^* - x \tag{24}$$

where, $x^*$, $x$ are the reference and the actual of the state vectors, and they are identified as follows:

$$\begin{cases} x^* = \begin{bmatrix} \Omega^* & i_d^* & i_q^* \end{bmatrix}^T \\ x = \begin{bmatrix} \Omega & i_d & i_q \end{bmatrix}^T \\ S(x,t) = \begin{bmatrix} S_\Omega & S_d & S_q \end{bmatrix}^T \end{cases} \tag{25}$$

The sliding surfaces for the speed and current loops of the PMSM are designed as follows:

$$\begin{bmatrix} S_\Omega \\ S_d \\ S_q \end{bmatrix} = \begin{bmatrix} \Omega^* - \Omega \\ i^* - i_d \\ i_q^* - i_q \end{bmatrix} \tag{26}$$

The derivative of the sliding surfaces-based PMSM model is calculated as follows:

$$\frac{d}{dt}\begin{bmatrix} S_\Omega \\ S_d \\ S_q \end{bmatrix} = \begin{bmatrix} \dot{\Omega}^* + \frac{F}{j}.\Omega - \frac{3.p}{2.j}.\psi_f.i_{q-eq}^* + \frac{1}{j}.T_l \\ \ddot{i_d}^* + \frac{R_s}{L_d}i_d - \frac{L_q}{L_d}p.i_q.\Omega - \frac{1}{L_d}v_{d-eq} \\ \ddot{i_q}^* + \frac{R_s}{L_q}i_q + \frac{L_d}{L_q}p.i_d.\Omega + p.\Omega\frac{\psi_f}{L_q} - \frac{1}{L_q}v_{q-eq} \end{bmatrix} \tag{27}$$

## 4.2. Determination of sliding conditions

The conditions that ensure the system dynamics converge towards the sliding surface and remain there despite disturbances are known as the conditions of existence and convergence. These conditions are categorized into two main types [52]:

- Direct approach: The earliest method, proposed and studied by Emelyanov [53] and Utkin [54], requires incorporating the values immediately before and after the commutation point for both S(x) and its derivative. This approach ensures the system's robustness and stability during the switching process. The condition is specified as follows:

$$\dot{S}(x).S(x) < 0 \tag{28}$$

- **Lyapunov approach:** In the Lyapunov approach, a Lyapunov candidate function $V(x) > 0$, a positive scalar function is selected for the system's state variables. The goal is to design a control law that ensures the derivative of this function, $\dot{V}(x) < 0$, is negative, thus guaranteeing system stability. The candidate Lyapunov function is constructed as follows [55,56]:

$$V(x) = \frac{1}{2}S^2(x) \tag{29}$$

By deriving the latter, we obtain:

$$\dot{V}(x) = \dot{S}(x).S(x) \tag{30}$$

To ensure that the function V(x) decreases, it is sufficient to guarantee that its derivative is negative. Thus, the requirement for convergence is stated as follows:Top of Form

$$\dot{S}.S < 0 \tag{31}$$

This condition is used to assess the robustness, stability, and control performance of nonlinear systems [57].

## 4.3 Calculation of control law

The control law in SMC consists of two components: the equivalent component $u_{eq}(t)$ and the switching component$u_s(t)$. It is formulated as follows:Top of Form

$$u(t) = u_{eq}(t) + u_s(t) \tag{32}$$

For the speed and current loop of the PMSM, $u_{eq}(t)$ *and* $u_s(t)$ are designed in the following vectors:

- Equivalent component $u_{eq}(t) = \begin{bmatrix} i_{q-eq}{}^* & V_{d-eq} & V_{q-eq} \end{bmatrix}^T$;

- Switching component $u_s(t) = \begin{bmatrix} i_{q-s}{}^* & V_{d-s} & V_{q-s} \end{bmatrix}^T$.

To derive the control law, the SMC operation typically unfolds through two main stages: the sliding phase and the approaching phase. The initial step, the sliding phase, revolves around selecting a suitable sliding manifold to maintain the system state on the desired trajectory until the desired control performance is attained. This phase involves computing the equivalent component $u_{eq}(t)$ based on the conditions outlined in Eq (33) to ensure the system remains stationary on the sliding surface.

$$\begin{cases} s(x, t) = 0 \\ \dot{s}(x, t) = 0 \end{cases} \tag{33}$$

By utilizing the sliding conditions inherent in the SMC design alongside Eq (33), the equivalent components of each PMSM loop can be identified as the ensuing vector:

$$\begin{bmatrix} i_{q-eq}{}^* \\ V_{d-eq} \\ V_{q-eq} \end{bmatrix} = \begin{bmatrix} \frac{2.j}{3.p.\psi_f}(\dot{\Omega}^* + \frac{F}{j}\Omega + \frac{1}{j}.T_l) \\ L_d.\dot{i_d}^* + R_s.i_d - L_q.p.i_q.\Omega \\ L_q.\dot{i_q}^* + R_s.i_q + L_d.p.i_d.\Omega + p.\Omega.\psi_f \end{bmatrix} \tag{34}$$

In the second phase, the system is compelled to reach the sliding surface within a finite time, rendering the system state an invariant manifold. This enables the formulation of an adequate control law, termed as the switching or reaching component $u_s(t)$. Hence, the requisite sliding conditions for this phase are delineated as follows [53,58]:

Top of Form

$$\begin{cases} s(x, t) = 0 \\ \dot{s}(x, t) \neq 0 \end{cases} \tag{35}$$

Based on these conditions and Eq (33), the switching term vector for the PMSM is formulated in Eq (33) wherein it is developed through a combination of the pseudo-sliding mode utilizing a continuous approximation (CA) technique and the ARL component grounded in the Exponential Reaching Law (ERL). Fig 5 depicts the nonlinear system using SMC scheme.

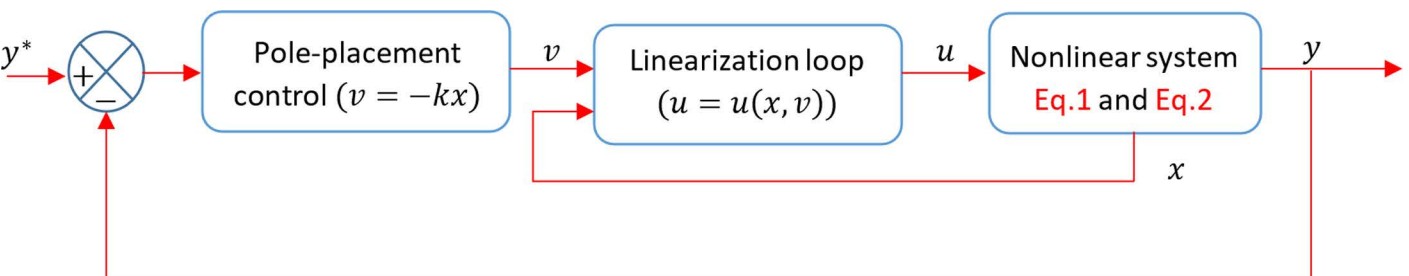

**Fig 5. Block diagram of SMC based nonlinear plant.**

$$\begin{bmatrix} i_{q-s}^* \\ v_{d-s} \\ v_{q-s} \end{bmatrix} = \begin{bmatrix} -\in_1 .sgn\,(S_\Omega) - k_1.S_\Omega \\ -\in_2 .sgn\,(S_d) - k_2.S_d \\ -\in_3 .sgn\,(S_q) - k_3.S_q \end{bmatrix} \tag{36}$$

## 5. Interpretation of Simulation Findings and Discussion

In this study, the effectiveness of the proposed control design and observation scheme was demonstrated using MATLAB SIMULINK. The control strategy employed a hybrid approach for the speed and current loops of the PMSM, integrating pseudo-SMC with an ARL. This technique was implemented using a CA technique to ensure smooth control action, while the ARL component was formulated based on the ERL, providing a robust convergence to the desired states. For sensorless speed observation, a MRAS was applied, enabling accurate speed estimation without the need for mechanical sensors. The simulation was conducted under various scenarios to assess the performance of the control and observation system thoroughly. These scenarios included step changes in speed reference, load torque disturbances, and parameter variations. Through this goal, the simulation of the sensorless speed of the PMSM based on the MRAS with the control design mentioned above is shown in Fig 6.

In test case 1, various speed levels (1432 rpm, 2864 rpm, 4297 rpm, and 5730 rpm) were applied under a load torque of 1 N.m to demonstrate the effectiveness of the ERL-SMC control and MRAS observation design. Despite the variation in speed levels, the control-observation scheme provided superior performance characteristics, such as a faster rise time (Tr) and minimal steady-state error (SSE). Specifically, as shown in Fig 7, the SSE for Level 1 (1432 rpm), Level 2 (2864 rpm), Level 3 (4297 rpm), and Level 4 (5730 rpm) ranged from 0.07% to 0.26%. The rise time (Tr) for Level 1

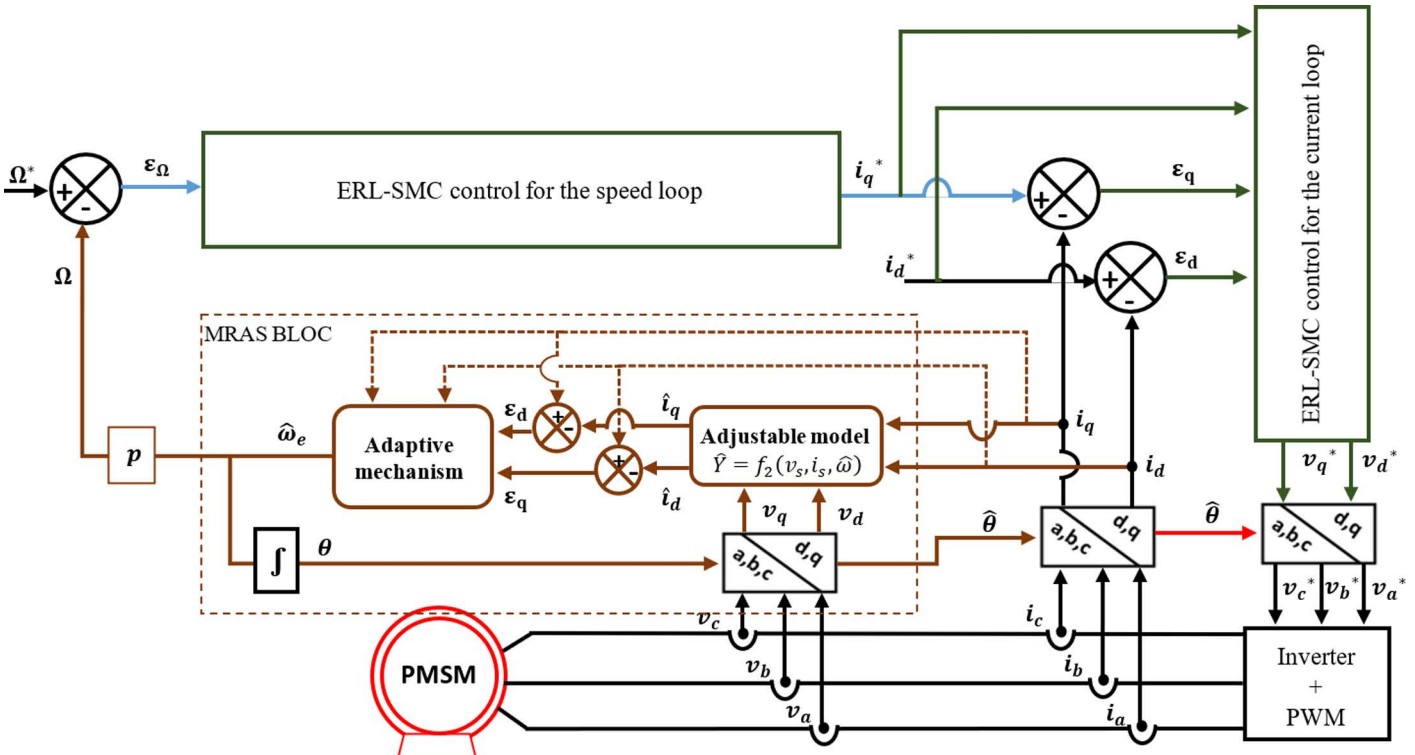

**Fig 6. PMSM drive based on the MRAS for the sensorless speed.**

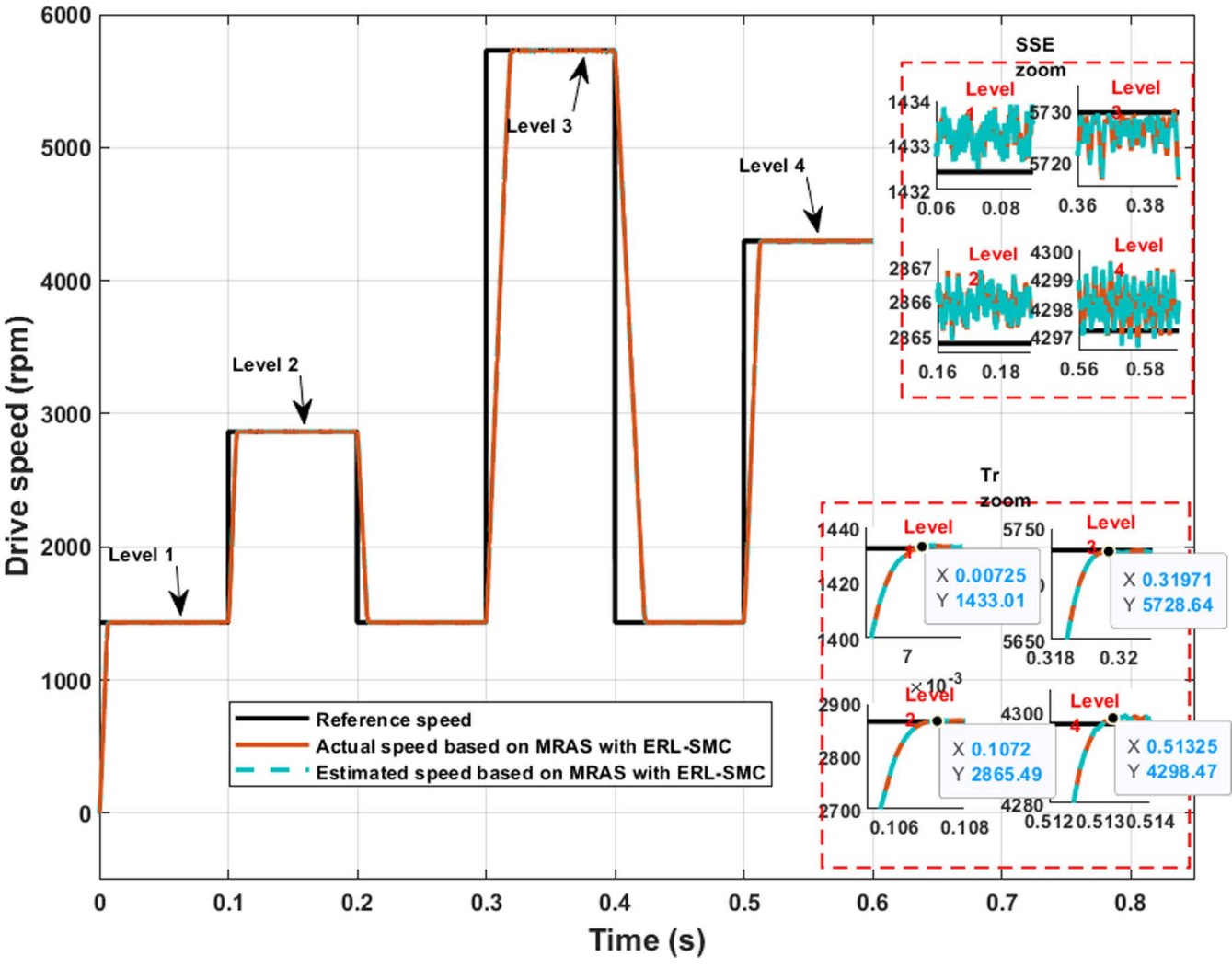

**Fig 7. PMSM Speed performance during Test case 1 (Speed levels change).**

and Level 2 remained stable at 7.2 ms and 10.8 ms, indicating a consistent and rapid response at these speed levels. However, for higher speed levels, the rise time slightly increased: for Level 3, it was recorded at 19.7 ms, and for Level 4, it reached 13.3 ms. Despite this increase, these values still represent a faster response time than typical high-speed level responses. Additionally, the control-observation scheme ensured that the actual and estimated speeds were equal, demonstrating its effectiveness. As detailed in Table 1, these results demonstrate the improved performance metrics achieved with the proposed control-observation scheme, highlighting its robustness and effectiveness across different operational speeds.

In the same test case, the proposed control-observation scheme also contributed to improved electromagnetic torque performance. This improvement was evident through the reduced ripple torque and lower overshoots during speed variations. The control-observation design effectively minimized torque ripple, resulting in smoother and more stable torque output, which is critical for maintaining the efficiency and reliability of the system. Additionally, the scheme's ability to limit overshoots during speed transitions further highlights its robustness, ensuring that the motor operates within desired parameters without experiencing significant deviations. These enhancements in torque characteristics not only improve

**Table 1. Detailed performance characteristics of PMSM under Test Cases 1, 2, 3, and 4.**

| Performance Characteristics | | SSE (%) | Tr (ms) | Undershoots/ overshoots (%) | Stability |
|---|---|---|---|---|---|
| During speed levels | Level 1 | 0.07 | 7.2 | 0 | incredibly |
| | Level 2 | 0.07 | 10.8 | 0 | incredibly |
| | Level 3 | 0.26 | 19.7 | 0 | incredibly |
| | Level 4 | 0.07 | 13.3 | 0 | incredibly |
| | 2290 rpm | 0.07 | 10.8 | 0 | incredibly |
| | Reverse speed | 0.07 | 10.8 | 0 | incredibly |
| | Rate of changing | 0.1 | 12.1 | | |
| During Disturbance events | Test Case 2 | 1st change | 0.07 | 10.8 | 0.11 | Good |
| | | 2nd change | 0.07 | 10.8 | 0.04 | Good |
| | | 3rd change | 0.07 | 10.8 | 0.11 | Good |
| | Test Case 3 | Reverse speed | 0.07 | 10.8 | 0.06 | Good |
| | Rate of change | 0.07 | 10.8 | | |
| During uncertainty events | -5% | 0.024 | 10.2 | 0 | Good |
| | -10% | 0.035 | 9.8 | 0 | Good |
| | 0% | 0.07 | 10.8 | 0 | Good |
| | +5% | 0.039 | 11.2 | 0 | Good |
| | +10% | 0.022 | 11.7 | 0 | Good |
| | Rate of change | 0.03 | 0.725 | | |

overall performance but also extend the lifespan of the motor by reducing mechanical stress and potential wear. The results, as illustrated in Fig 8, confirm the effectiveness of the ERL-SMC control and MRAS observation design in achieving superior torque control across different operational speeds.

In the same test case, the proposed control-observation scheme demonstrated significant improvements in the current performance of the PMSM. The enhancements were particularly notable in the minimization of the tolerance band for both d-q currents and stator currents, as illustrated in Figs 9.a and 9.b, respectively. The d-q current control is crucial for the optimal operation of PMSMs. In this test case, the implementation of the ERL-SMC control and MRAS observation design resulted in a narrower tolerance band for the d-axis and q-axis currents. This precision in current control is essential for reducing current ripples, which directly impacts the overall efficiency and performance of the PMSM. The reduced tolerance band indicates that the currents are maintained closer to their reference values, thereby minimizing deviations and ensuring smoother operation. This is particularly important in applications requiring high precision and stability. Fig 9a shows the improved performance in the d-q currents. The control scheme effectively suppresses oscillations and maintains the currents within a tighter range around the desired set points. This reduction in current ripple leads to lower electromagnetic torque ripple, which enhances the smoothness of the motor's operation and reduces mechanical vibrations. As a result, the overall lifespan of the motor components is extended, and the noise generated by the motor is significantly reduced. Furthermore, the improvements are also evident in the stator currents, as depicted in Fig 9b. The control-observation design minimizes the fluctuations in the stator currents, maintaining them within a narrow tolerance band. This precise control of stator currents is vital for reducing losses due to harmonics and ensuring efficient energy conversion within the motor. The reduced stator current ripple contributes to lower thermal stress on the motor windings, preventing overheating and enhancing the motor's reliability and durability.

In Test Case 2, the robustness of the control-observation scheme was demonstrated by applying disturbances in the form of load torques of 1 N.m and 2 N.m at different instances while maintaining a constant speed of 2290 rpm. As

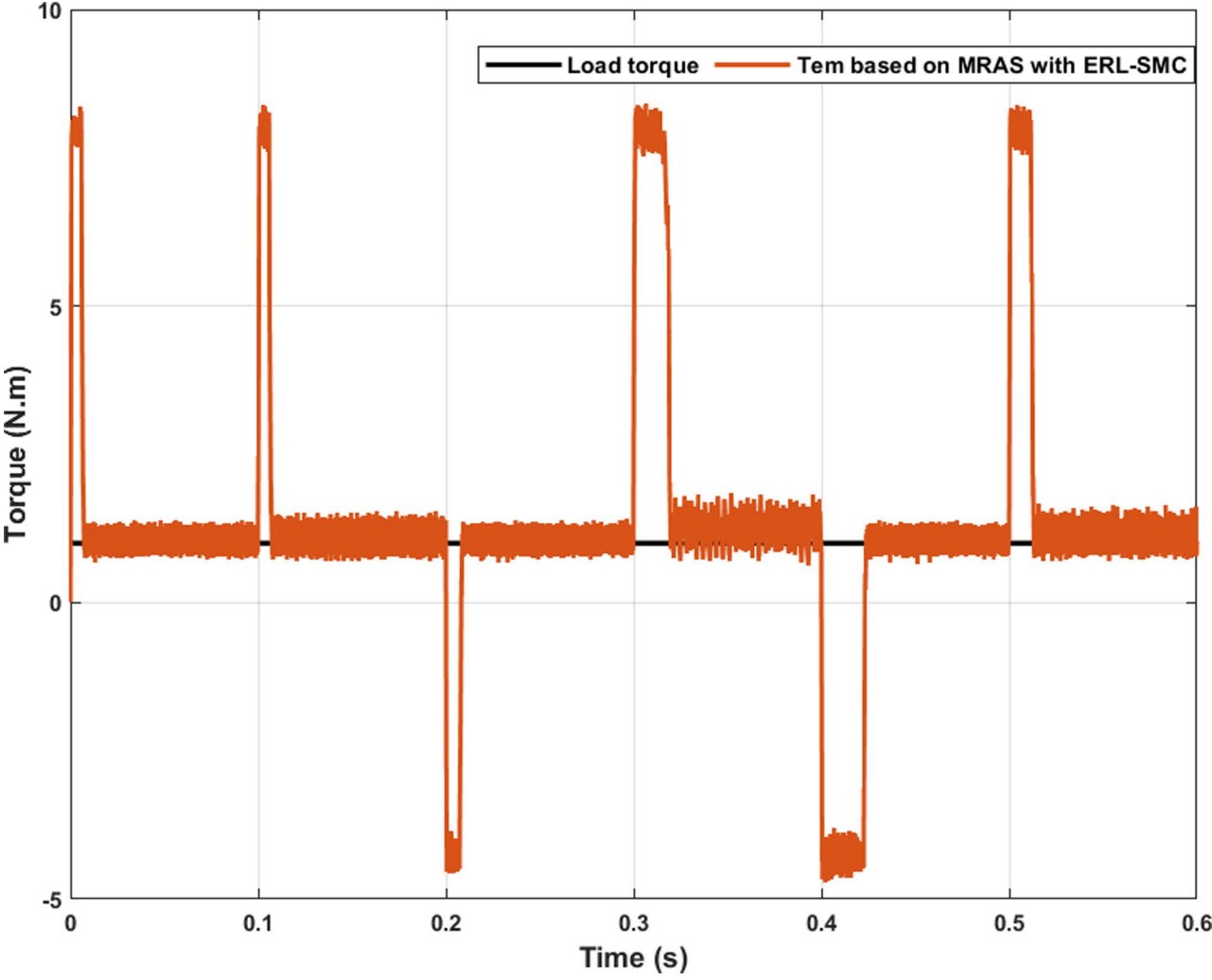

**Fig 8. PMSM Electromagnetic torque performance during Test case 1 (Speed levels change).**

shown in Fig 10, the speed performance of the PMSM was significantly enhanced under these disturbances. The control-observation scheme effectively maintained good speed characteristics despite the applied load torques. The rise time (Tr) was estimated to be 10.8 ms, indicating a rapid response compared to higher speeds. This quick rise time underscores the system's ability to promptly reach the desired speed, even under varying load conditions. Furthermore, undershoots and overshoots during the load torque changes were minimal, estimated between 0.04% and 0.11%. These small deviations highlight the robustness of the control-observation scheme in handling disturbances, ensuring that the motor's speed remains stable and within desired limits. Moreover, despite the disturbances applied, the SSE was kept exceptionally low, at approximately 0.07%. This indicates that the control-observation scheme can effectively stabilize speed performance, maintaining high precision and minimal error in steady-state conditions. Additionally, the control-observation scheme ensured that the actual and estimated speeds were equal, further demonstrating its effectiveness. These results demonstrate the superior performance of the proposed control-observation scheme in maintaining stability and precision under varying load conditions. The system's ability to quickly respond to changes and minimize errors underscores its robustness and reliability for high-performance motor control applications.

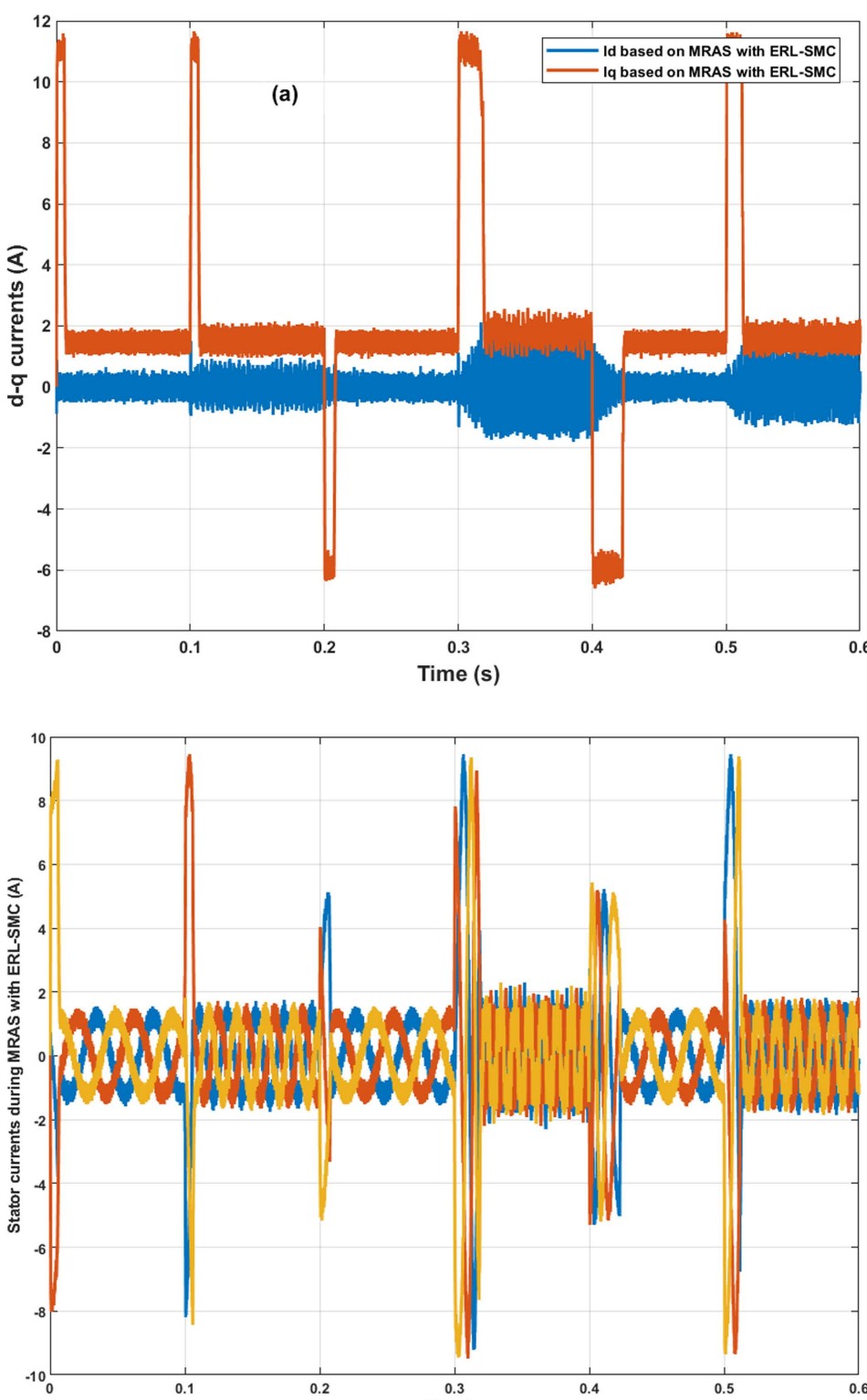

**Fig 9. PMSM Currents during Test case 1 (Speed levels change); (a): direct-quadratic currents; (b) stator currents.**

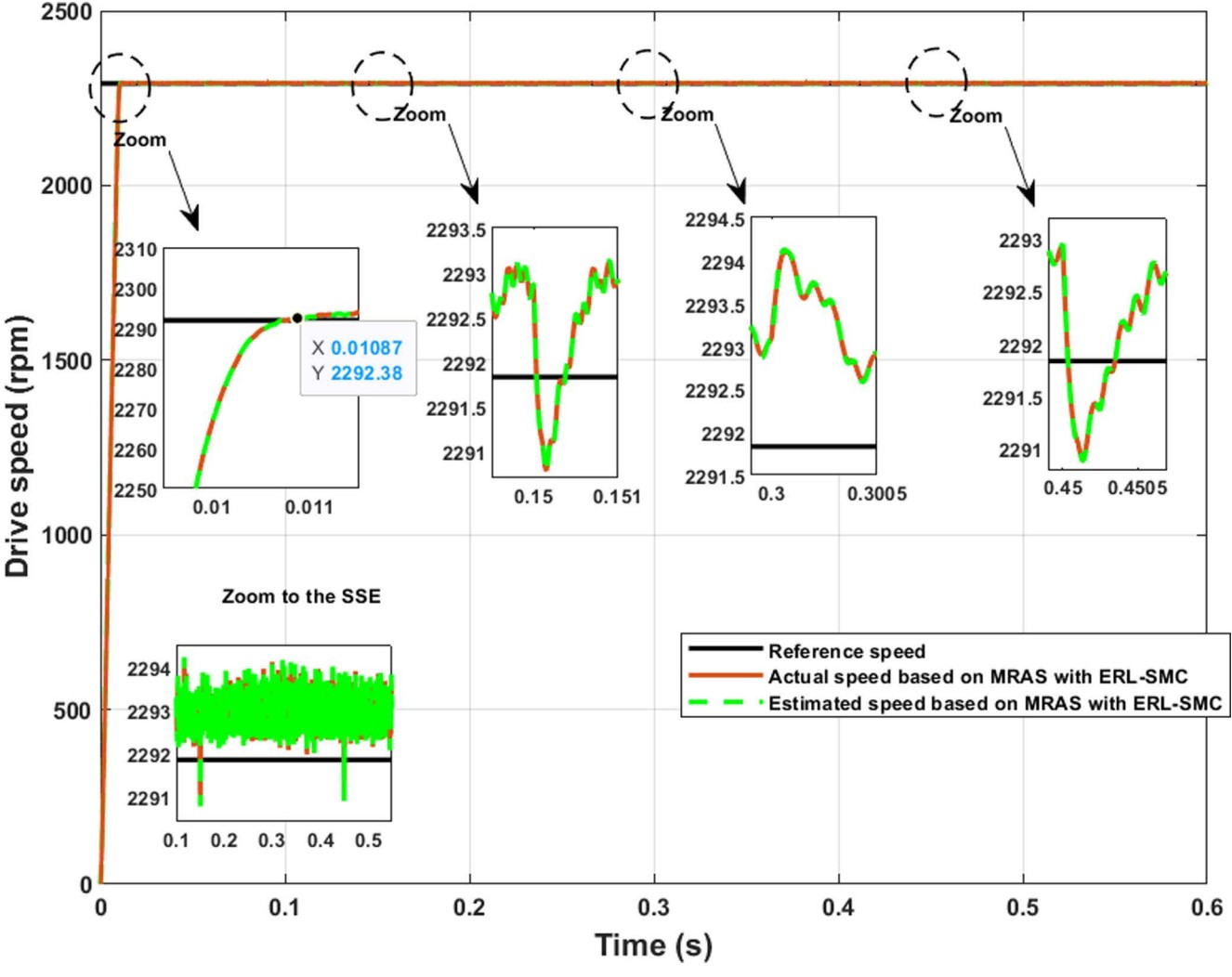

**Fig 10. PMSM Speed performance during Test case 2.**

Regarding electromagnetic torque performance, the control-observation scheme demonstrated exceptional effectiveness in maintaining stable torque characteristics. As shown in Fig 11, the electromagnetic torque did not exhibit initial overshoots when the load torque was altered. Furthermore, the torque ripple remained minimal even under high-speed conditions and applied disturbances. This stability in torque output is vital for ensuring smooth and reliable operation of the PMSM, as it reduces mechanical stress and enhances overall system durability. The consistent torque performance under varying conditions highlights the scheme's robustness and precision, underscoring its capability to manage high-speed and disturbances while maintaining superior electromagnetic performance.

In Test Case 2, the control-observation scheme showcased its effectiveness in enhancing the performance of the direct-quadrature (d-q) currents. As illustrated in Fig 12a, the precise control enabled by the scheme minimized the tolerance band of these currents, ensuring that they remained close to their reference values even under varying load torques and high-speed conditions. This tight regulation of the d-q currents is crucial for reducing torque ripples and ensuring smooth motor operation. Furthermore, the stator currents exhibited improved stability and accuracy, as illustrated in Fig 12b. The control-observation design effectively reduced fluctuations within a narrow range, improving energy conversion

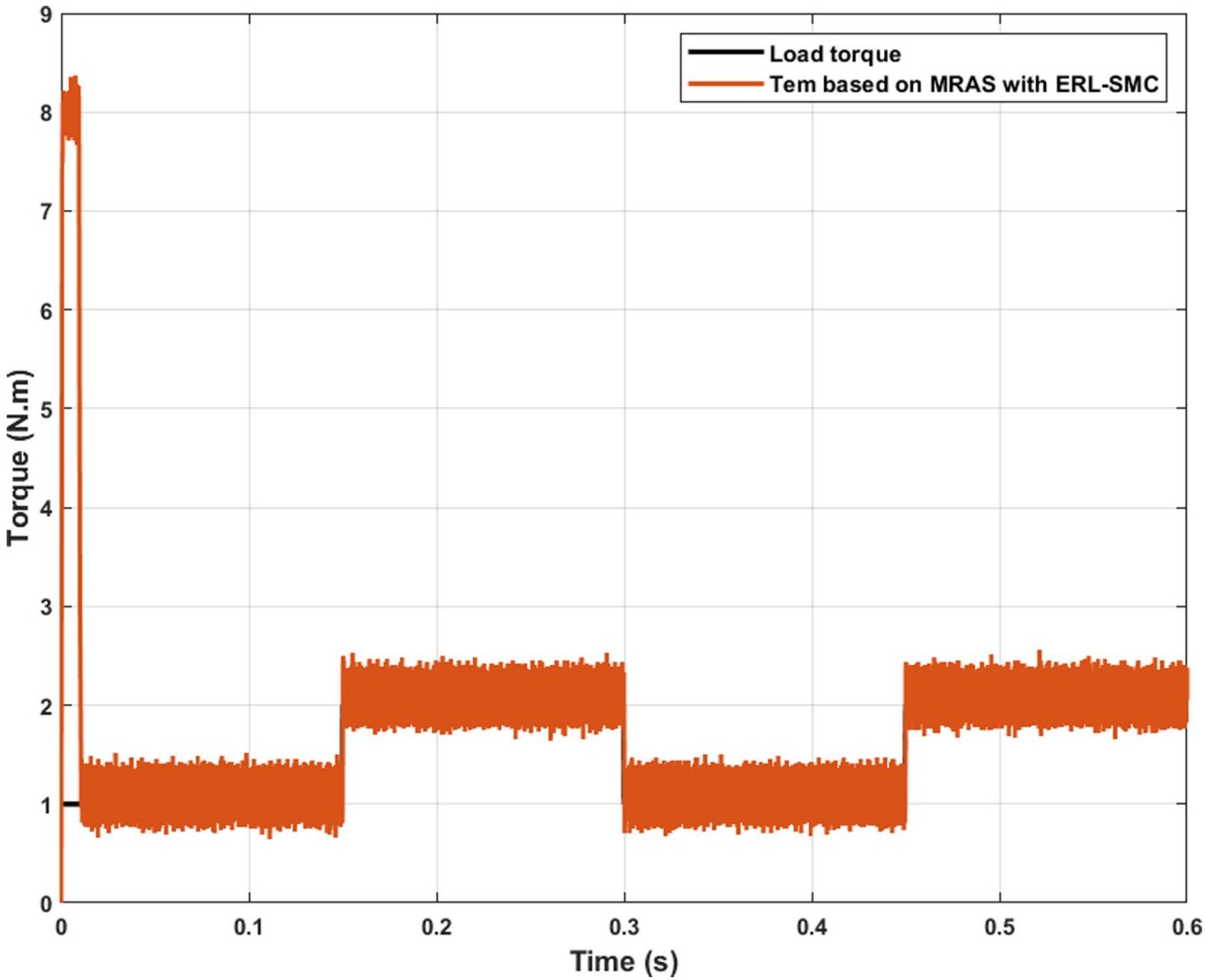

**Fig 11. PMSM Electromagnetic torque performance during Test case 2.**

efficiency and reducing harmonic losses. This stability in stator currents is particularly beneficial for maintaining consistent motor performance under dynamic load conditions and high-speed operations. The control-observation scheme also avoided overshoots and undershoots during transitions when the load torque changed. This capability ensured the currents did not experience significant deviations, maintaining a stable and reliable motor performance. The prevention of overshoots and undershoots is critical for reducing mechanical stress and enhancing the smoothness of the motor's response. Finally, the overall stability of the current performance under high-speed conditions and disturbances highlighted the robustness of the control-observation scheme. The ability to maintain precise control over both d-q and stator currents despite challenging operating conditions underscores the effectiveness of the design in delivering reliable and efficient motor control.

In Test Case 3, the control-observation scheme was tested under reverse rotation speed with applied disturbances. The PMSM speed varied from 2290 to -2290rpm within 0.2 s, and then back from -2290–2290 within 0.4 s. Simultaneously, the load torque was altered between 1 N.m and 2 N.m at instants of 0.1 s, 0.3 s, and 0.5 s, Fig 13. Under these test conditions, the control-observation scheme maintained its effectiveness in handling reverse speed applications and demonstrated

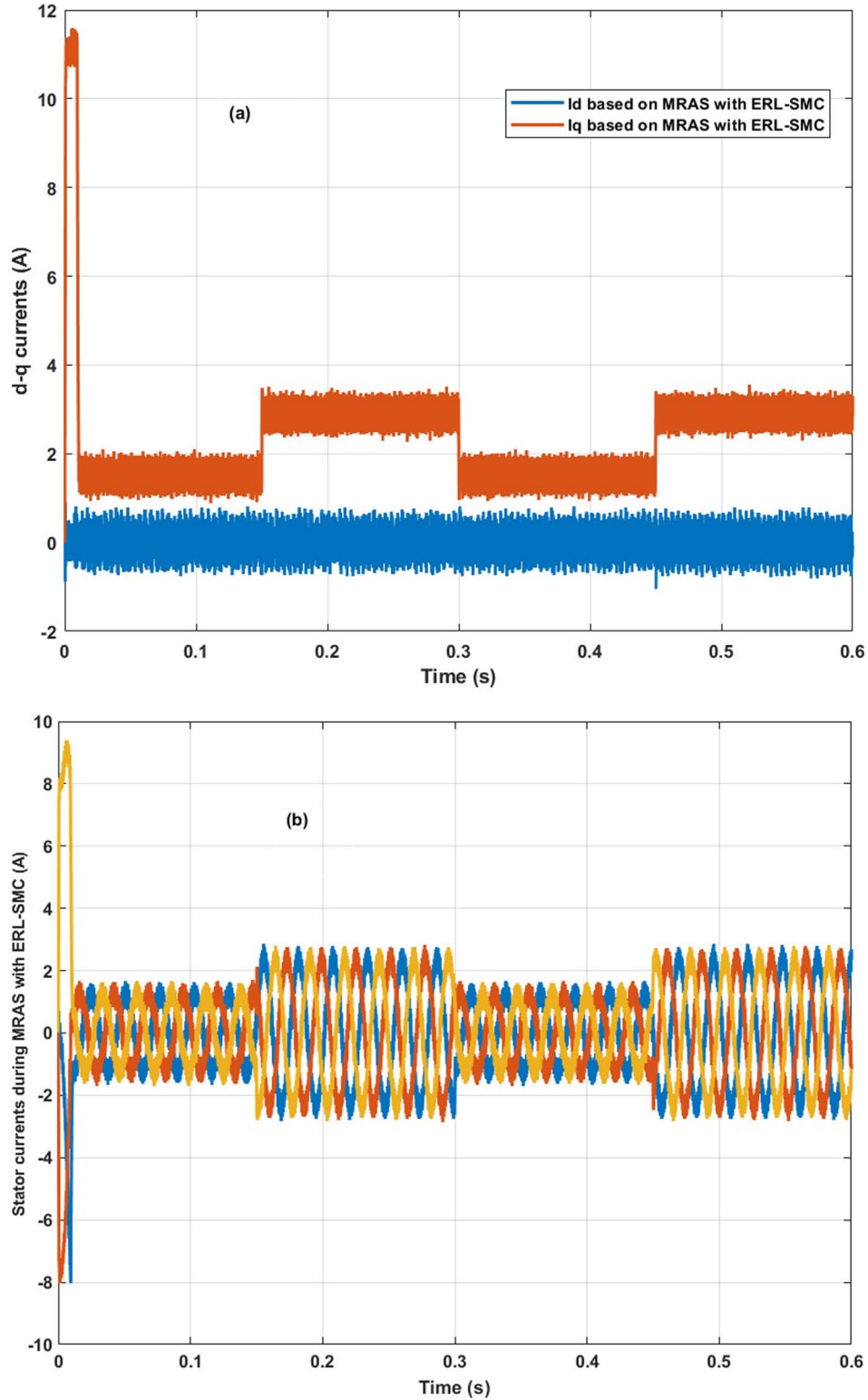

**Fig 12. PMSM Currents during Test case 2 (a): direct-quadratic currents; (b) stator currents.**

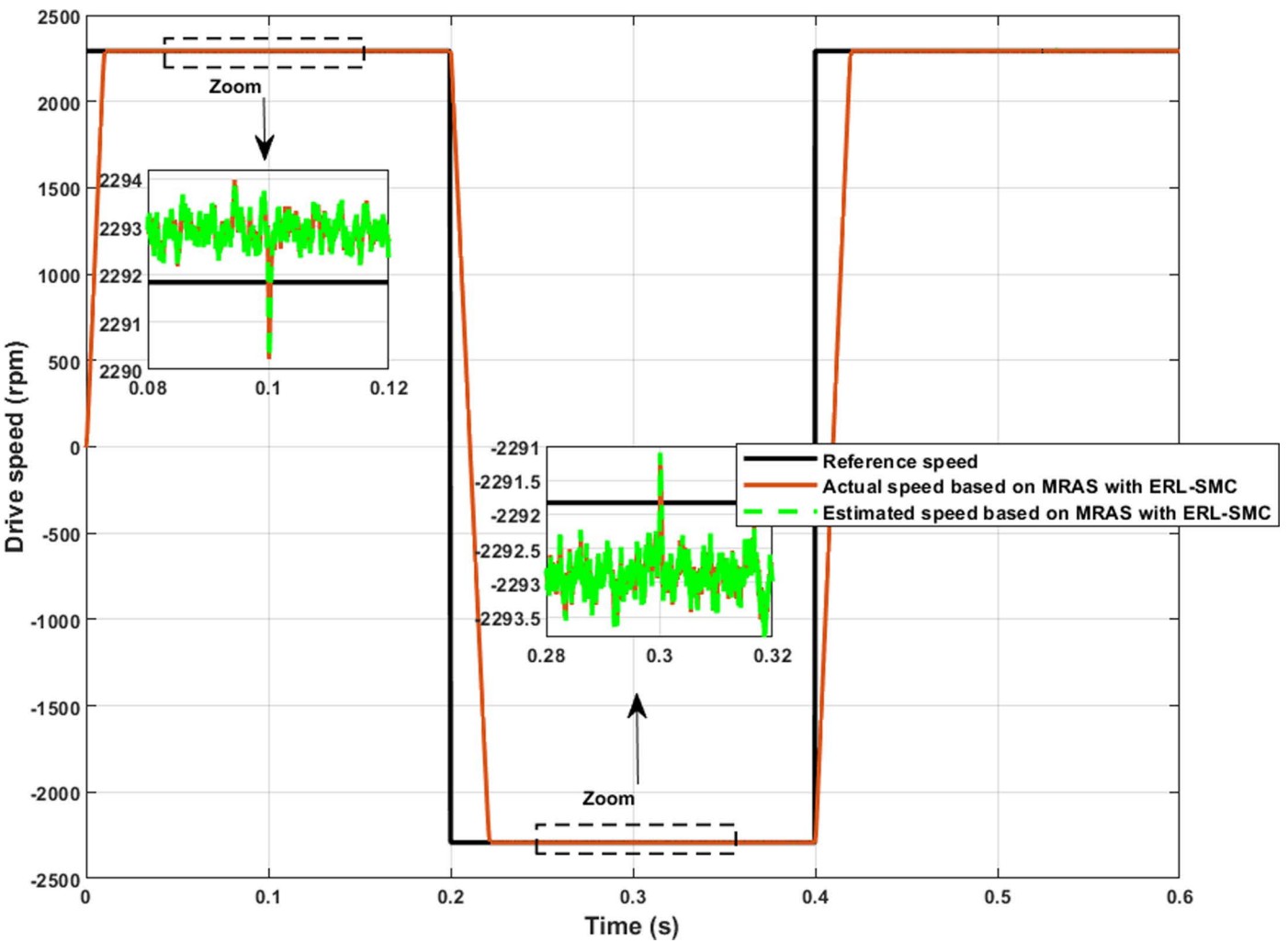

**Fig 13. PMSM Speed performance during Test case 3 (reversed rotation).**

robustness under varying load torques. Similar to the previous test cases, the scheme improved the characteristic performance of the PMSM speed. The SSE was remarkably low, estimated at 0.07%, even during reverse speed operation, highlighting the scheme's precision. Furthermore, the control-observation scheme effectively managed overshoots and undershoots during rotational and reverse rotational speed transitions. The overshoots and undershoots were estimated at 0.11% and 0.06%, respectively. These minimal deviations demonstrate the scheme's ability to maintain stable and reliable performance under challenging conditions, including reverse rotational speed and load disturbances. Additionally, the control-observation scheme ensured that the actual and estimated speeds were equal, further demonstrating its effectiveness. Test Case 3 confirmed the robustness and effectiveness of the control-observation scheme in handling complex speed variations and load changes, ensuring high performance and stability of the PMSM.

The control-observation scheme effectively managed the electromagnetic torque performance of the PMSM, demonstrating its robustness even under reverse rotational speed conditions. As depicted in Fig 14, the scheme successfully minimized torque ripple during reverse speed operation, similar to the performance observed in previous test cases. The results revealed that the scheme had maintained stability and effectively prevented initial overshoots when the load torque

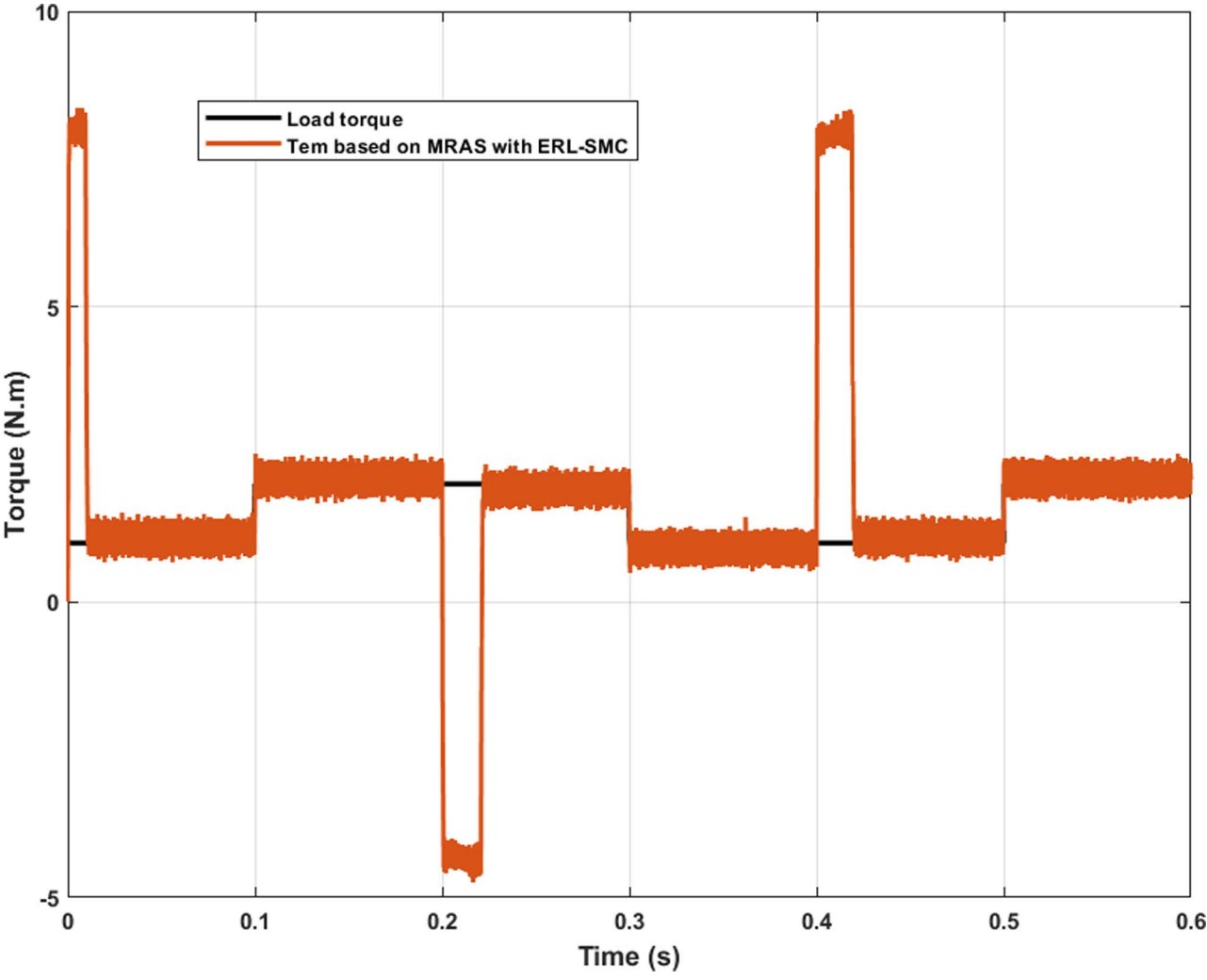

**Fig 14. PMSM Electromagnetic torque performance during Test case 3 (reversed rotation).**

was altered. This capability underscores the scheme's proficiency in handling dynamic load changes and reverses speed scenarios without compromising torque stability. The consistent performance, with minimized ripple torque and no significant overshoots, highlights the continued effectiveness of the control-observation scheme in ensuring smooth and reliable operation of the PMSM across various challenging conditions.

Figs 15a and 15b illustrate the enhanced performance of the direct-quadrature (d-q) currents and stator currents of the PMSM during reverse rotational speed and simultaneous load torque application. The control-observation scheme demonstrated its capability to improve current stability and reduce the tolerance band even under these challenging conditions. During reverse speed operation combined with load torque disturbances, the scheme maintained minimal deviations in both d-q and stator currents, ensuring efficient and reliable motor performance. This level of precise current control highlights the scheme's effectiveness in managing complex scenarios involving both dynamic speed changes and varying load torques, reinforcing its robustness and efficiency in diverse operational conditions.

In Test Case 4, uncertainties were introduced to further demonstrate the robustness of the control-observation scheme applied to the PMSM performance. As illustrated in Fig 16, the speed performance under this scenario was examined with

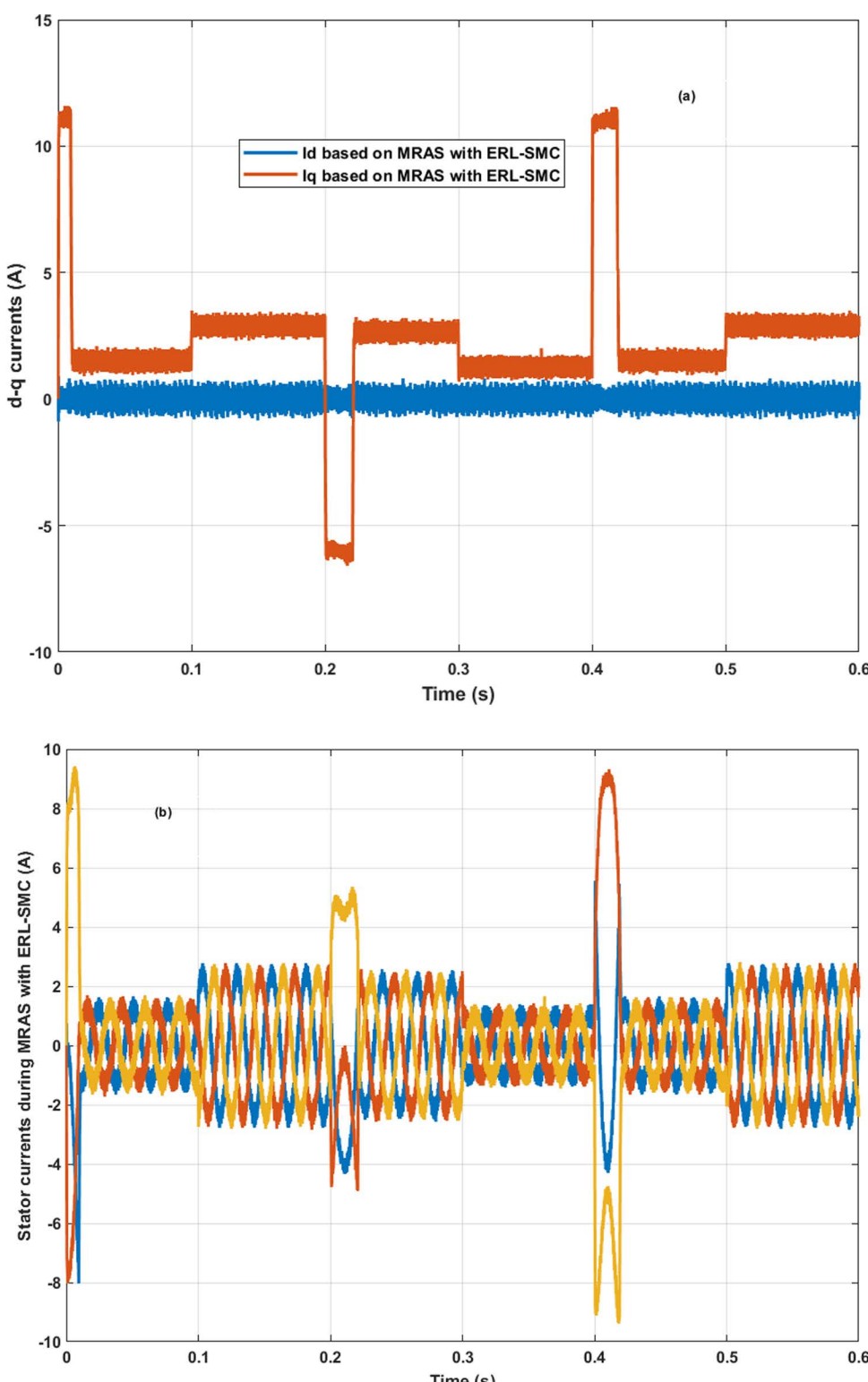

**Fig 15.** PMSM Currents during Test case 3 (reversed rotation); (a): direct-quadratic currents; (b) stator currents.

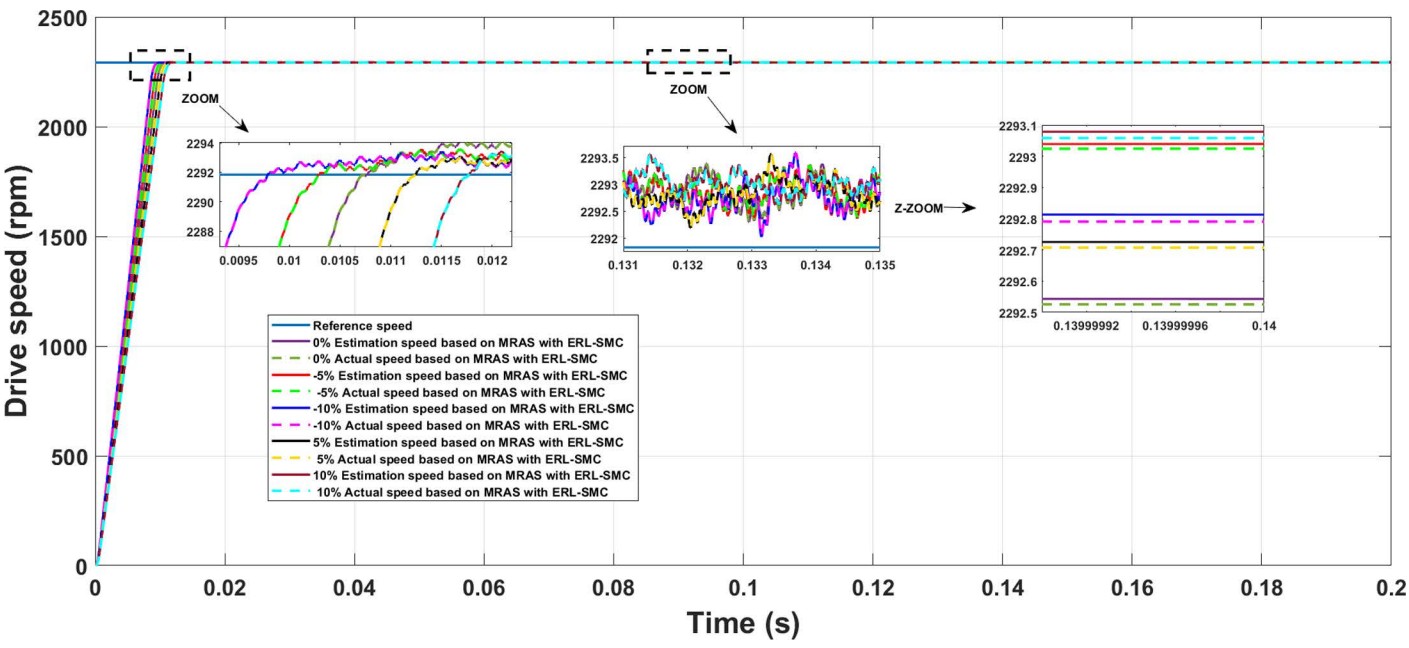

**Fig 16. PMSM Speed performance during Test case 3 (uncertainties events).**

PMSM parameters (resistance, inductances, and inertia) varying by -5%, -10%, 0%, +5%, and +10%, while maintaining a constant rotational speed of 2290 rpm and applying a load torque of 1 N.m. Despite these challenging conditions, the speed performance remained unaffected in terms of rise time and SSE, showcasing the effectiveness of the control-observation scheme. The variation in rise time was extremely low, estimated between 9.5 ms and 11.7 ms, indicating a rate of change of only 0.725 ms. This minimal fluctuation demonstrates the system's ability to maintain a consistent and rapid response, even when subjected to parameter uncertainties. Additionally, the control-observation scheme ensured that the actual and estimated speeds were identical, highlighting its precision and reliability. Similarly, the SSE showed a negligible percentage change, estimated between 0.03% and 0.05%, with a rate of change of only 0.03%. This minimal deviation underscores the scheme's capacity to maintain high accuracy and minimal error in steady-state conditions, even under varying parameter conditions. The alignment between the estimated and actual speeds reinforces the robustness of the control-observation scheme in maintaining optimal performance despite uncertainties. These results validate the superior performance of the proposed control-observation scheme in managing uncertainties and ensuring stable and precise speed control of the PMSM. Maintaining consistent rise times and minimal SSE under varying conditions underscores the scheme's robustness and suitability for high-performance motor control applications, ensuring reliability and efficiency in real-world scenarios.

The comparative scenario was conducted to evaluate the performance of three schemes: the control-observation (MRAS with ERL-SMC), classical ERL-SMC (Type 1) using sensor speed, ERL-SMC based on the pseudo-sliding mode (Type 2) using sensor speed and FOC using sensor speed. The test conditions involved maintaining a speed of 2290 rpm and applying a load torque variation from 2 N.m to 5 N.m at 0.2 s. Fig 17 illustrates the speed performance under these conditions. The control-observation scheme (MRAS with ERL-SMC) demonstrated superior performance and robustness compared to both Type 1, Type 2 control and FOC schemes. This superiority is evident in the key performance metrics and the scheme's ability to handle disturbances effectively. According to Table 2, the rise time for the control-observation scheme was estimated to be 10.8 ms, significantly faster than the 14 ms, 16 ms and 26 ms rise times observed for the Type

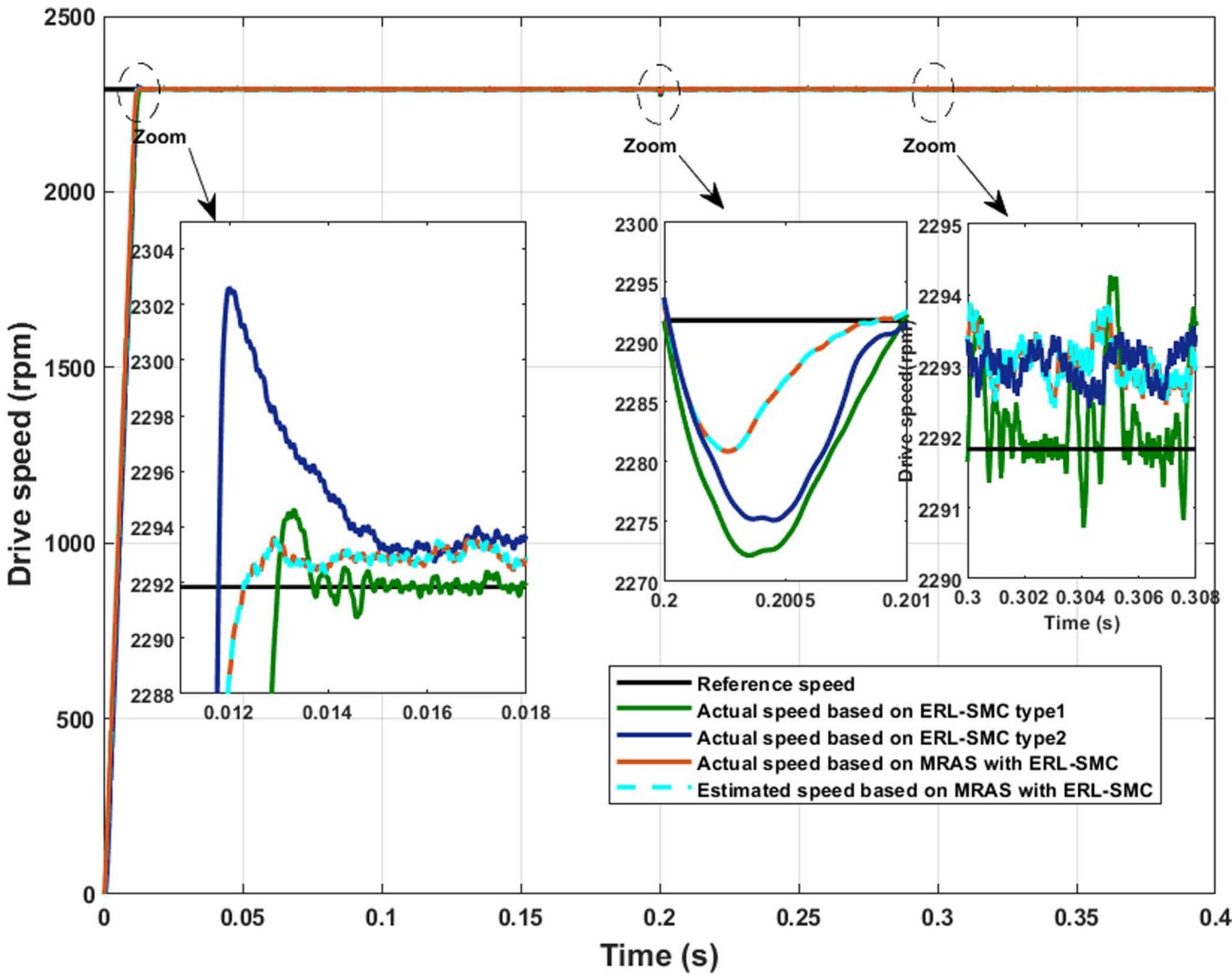

**Fig 17. PMSM speed performance, Test case 4.**

**Table 2. Comparative Performance Analysis of PMSM in Test Cases 4 and 5.**

| Applied scheme | MRAS with ERL-SMC | ERL-SMC (Type 1) | ERL-SMC (Type 2) | FOC |
|---|---|---|---|---|
| $T_r$ (ms) | 10.8 | 14 | 16 | 26 |
| SSE (%) | 0.07 | 0.13 | 0.07 | 0.14 |
| Speed Undershoots/ overshoots (%) | 0.41 | 0.87 | 0.74 | 1.31 |
| Initial speed overshoots (%) | 0 | 0.4 | 0.09 | 0.13 |
| Initial $T_{em}$ overshoots when the $T_l$ changed (%) | 24 | 30 | 46 | 60 |
| Currents tolerance band | +0.488 −0.246 | +0.696 −0.349 | +0.477 −0.276 | +1.363 −0.909 |
| THD (%) | 17.55 | 27.7 | 19.79 | 34.74 |

1 and Type 2 schemes, respectively. Notably, the control-observation scheme exhibited no initial overshoots, a significant advantage over the Type 1, Type 2 and FOC schemes, which both suffered from initial overshoots. This characteristic underscores the stability and precision of the control-observation approach in maintaining the desired speed without transient deviations. Regarding SSE, the control-observation scheme achieved the lowest SSE among the four schemes, indicating higher accuracy and stability in maintaining the target speed. This result reflects the scheme's enhanced capability in minimizing errors and ensuring consistent speed performance, even under varying load conditions. During the application of load torque, the control-observation scheme recorded the smallest undershoots, estimated at 0.48%. This minimal deviation further highlights the robustness of the control-observation scheme in handling sudden changes in load without significant performance degradation. Additionally, the control-observation scheme effectively minimized chattering, contributing to smoother and more stable speed performance compared to the Type 1, Type 2 and FOC. Furthermore, the control-observation scheme provided superior rise time and minimal SSE and ensured that the actual and estimated speeds were identical, reinforcing its precision and reliability. These results confirm the control-observation scheme's efficacy in delivering high-performance motor control, outperforming the classical, pseudo-sliding mode ERL-SMC and FOC schemes under the specified test conditions.

In the same scenario, the electromagnetic torque performance of the PMSM under the control-observation scheme (MRAS with ERL-SMC) proved to be superior compared to the classical ERL-SMC (Type 1), the pseudo-sliding mode ERL-SMC (Type 2) and FOC schemes as in Fig 18. The control-observation scheme exhibited the smallest initial overshoot when the load torque varied from 2 N.m to 5 N.m, with the overshoot estimated at just 1.25 N.m. This minimized overshoot demonstrates the scheme's capability to handle sudden load changes with greater stability and precision. Furthermore, the control-observation scheme also achieved the lowest ripple torque among the three schemes. This lower ripple torque indicates a more consistent and stable electromagnetic torque output, which is crucial for maintaining smooth motor operation and reducing mechanical stress. The superior performance in minimizing both initial overshoots and ripple torque highlights the effectiveness of the control-observation scheme in providing robust and reliable motor control under varying load conditions. These results underline the control-observation scheme's enhanced capability in delivering precise and stable electromagnetic torque performance, ensuring optimal operation of the PMSM even in challenging scenarios with significant load variations.

The direct and quadratic current performances of the PMSM under the control-observation scheme (MRAS with ERL-SMC) showed marked improvements compared to the classical ERL-SMC (Type 1), the pseudo-sliding mode ERL-SMC (Type 2) and FOC schemes. As illustrated in Fig 19, the control-observation scheme achieved the smallest tolerance band for both d-q currents. This reduced tolerance band signifies more precise and stable current control, crucial for efficient motor operation and minimizing electrical noise. The control-observation scheme's ability to maintain consistent current levels, despite varying load conditions, highlights its precision in current control. This precision ensures that the PMSM operates smoothly, with fewer fluctuations and enhanced overall stability. The superior performance of the control-observation scheme in managing d-q currents underscores its robustness and efficacy in providing high-quality motor control.

The stator current performance of the PMSM under the control-observation scheme (MRAS with ERL-SMC) demonstrated remarkable improvements in stability, minimal overshoots, and a reduced tolerance band when compared to the classical ERL-SMC (Type 1), pseudo-sliding mode ERL-SMC (Type 2) and FOC schemes. This is clearly illustrated in Fig 20. The control-observation scheme's superior stability in stator current ensures a more consistent and efficient operation of the PMSM, which is crucial for minimizing electrical noise and enhancing the overall system's performance. The significantly lower overshoots observed with the control-observation scheme indicate its effectiveness in maintaining precise current levels, even during dynamic conditions and load changes. This precise control reduces the risk of damaging the motor and contributes to smoother operation. Additionally, the smallest tolerance band achieved by the control-observation scheme highlights its capability to provide accurate current control, minimizing deviations from the desired current values. This accuracy is essential for optimizing motor performance, reducing energy losses, and extending the lifespan of the motor components.

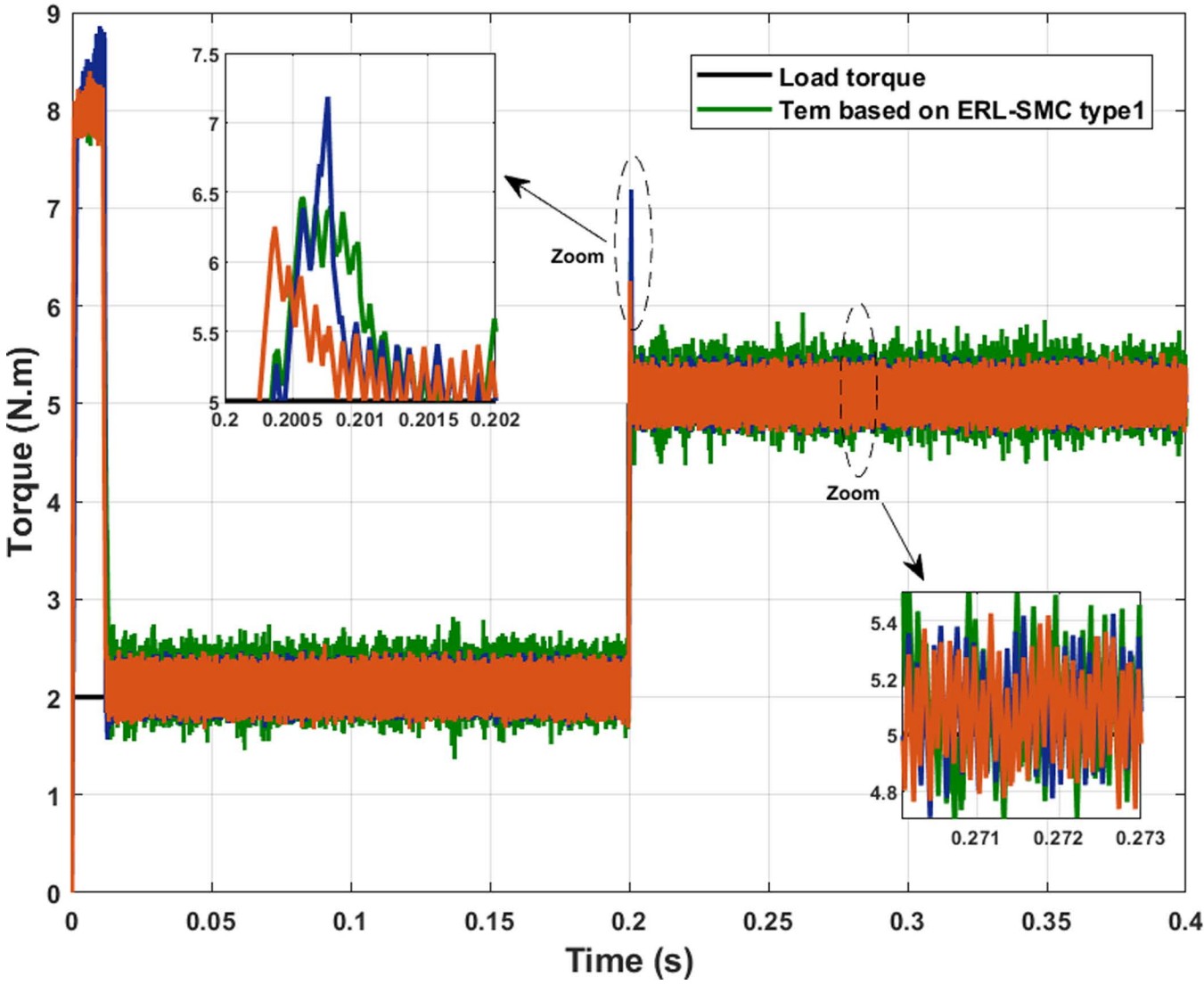

**Fig 18. PMSM electromagnetic torque performance PMSM during Test case 4.**

In the comparative study of harmonic analysis of current, the Total Harmonic Distortion (THD) values for the three control schemes were analyzed to assess their effectiveness. The control-observation scheme (MRAS with ERL-SMC) exhibited the lowest THD at 17.55% as depicted in Fig 21a, indicating superior performance in minimizing harmonic distortion in the current. This low THD value signifies a cleaner and more efficient current waveform, which is beneficial for reducing electrical noise and improving the overall efficiency of the PMSM system. In contrast, the classical ERL-SMC (Type 1) and FOC schemes showed the highest THD at 27.70% and 34.74% as shown in Figs 21b and 21d, reflecting significant harmonic content and potential issues with current distortion. The pseudo-sliding mode ERL-SMC (Type 2) scheme had a THD of 19.79%, Fig 21c, which, while better than Type 1, still lagged behind the control-observation scheme. These results underscore the effectiveness of the control-observation scheme in delivering a more refined and harmonically pure current, making it the most suitable choice for applications requiring high precision and efficiency.

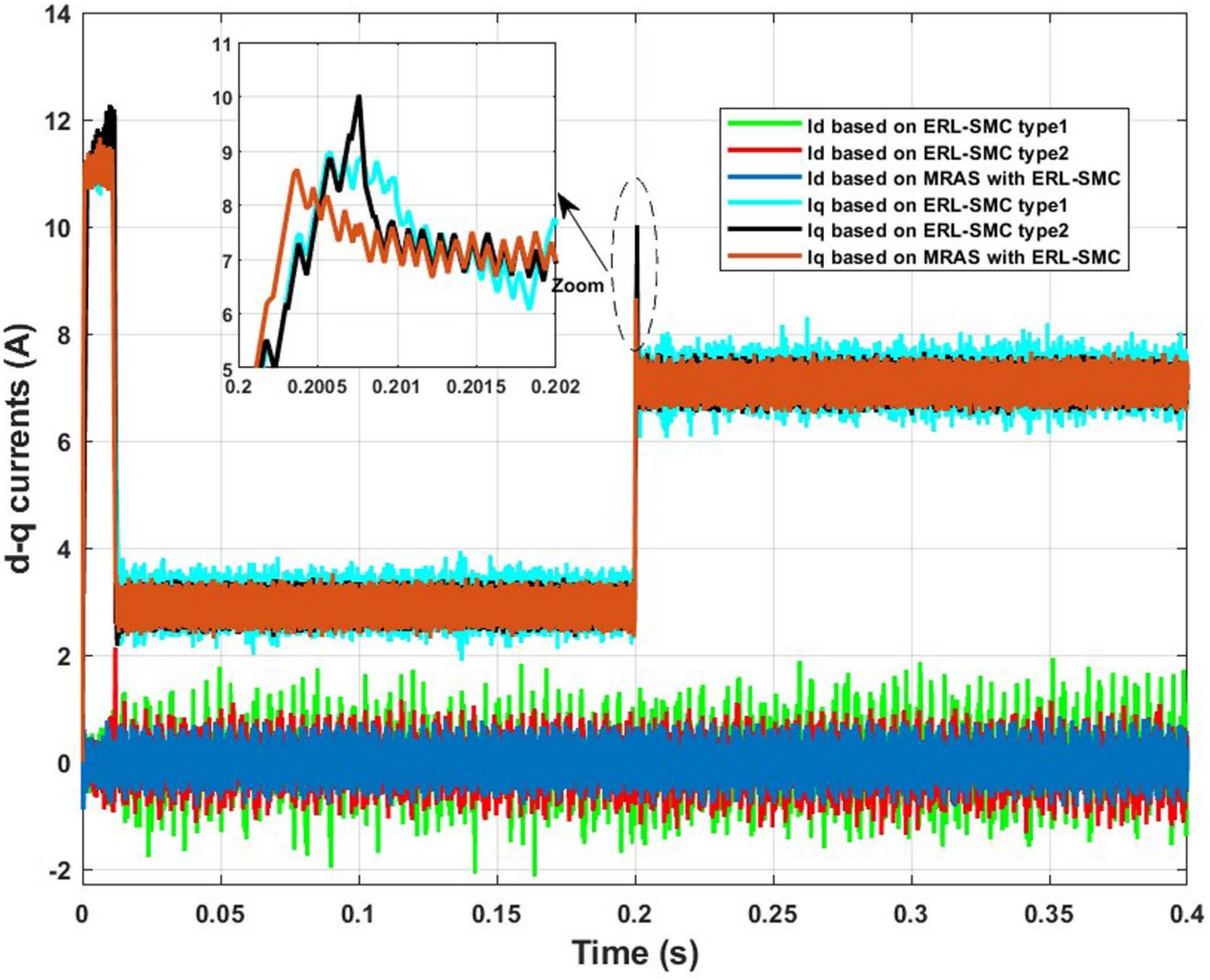

**Fig 19. PMSM direct and quadratic currents during Test case 4.**

The various test cases and scenarios applied throughout the simulations unequivocally demonstrate the effectiveness and robustness of the control-observation scheme (MRAS with ERL-SMC). Across different speed levels, load torques, and even under the presence of disturbances and reverse rotation, the scheme consistently maintained superior performance characteristics. The control-observation scheme exhibited rapid rise times, minimal SSEs, and significantly reduced overshoots and undershoots, ensuring stable and reliable PMSM operation. Furthermore, the scheme's ability to minimize current ripple, maintain a tight tolerance band for both d-q and stator currents, and achieve low THD values underscores its capability to deliver high precision and efficiency. Additionally, the advantages of sensorless speed control, such as reduced hardware complexity, lower maintenance requirements, and improved reliability in harsh environments, further enhance the scheme's practicality and cost-effectiveness. These benefits eliminate the need for physical sensors, which are often prone to failure and add to the overall system cost and complexity. The sensorless approach ensures a more streamlined and robust solution, capable of maintaining optimal performance even in adverse conditions. These

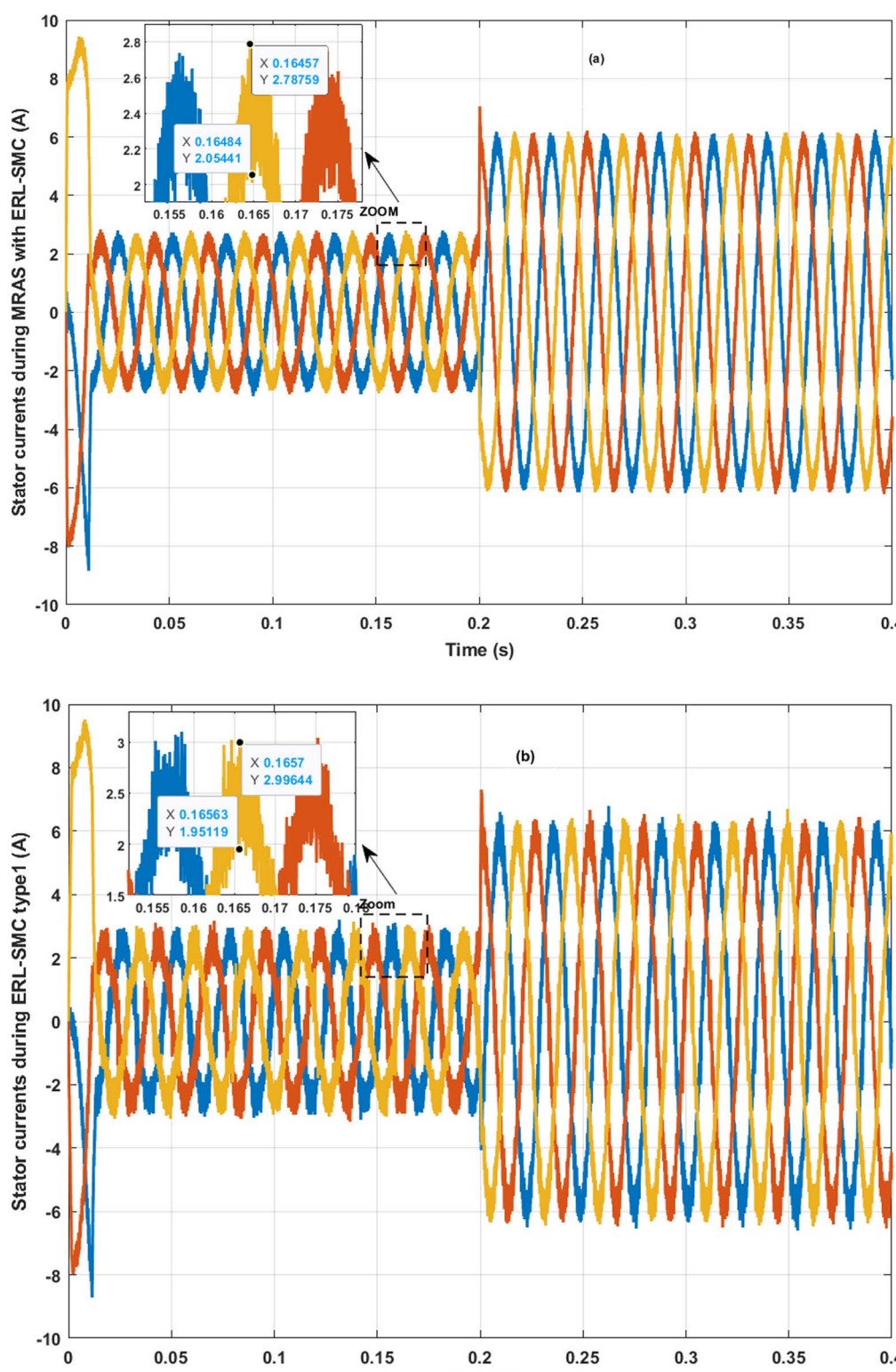

**Fig 20. PMSM Stator currents during Test case 4.**

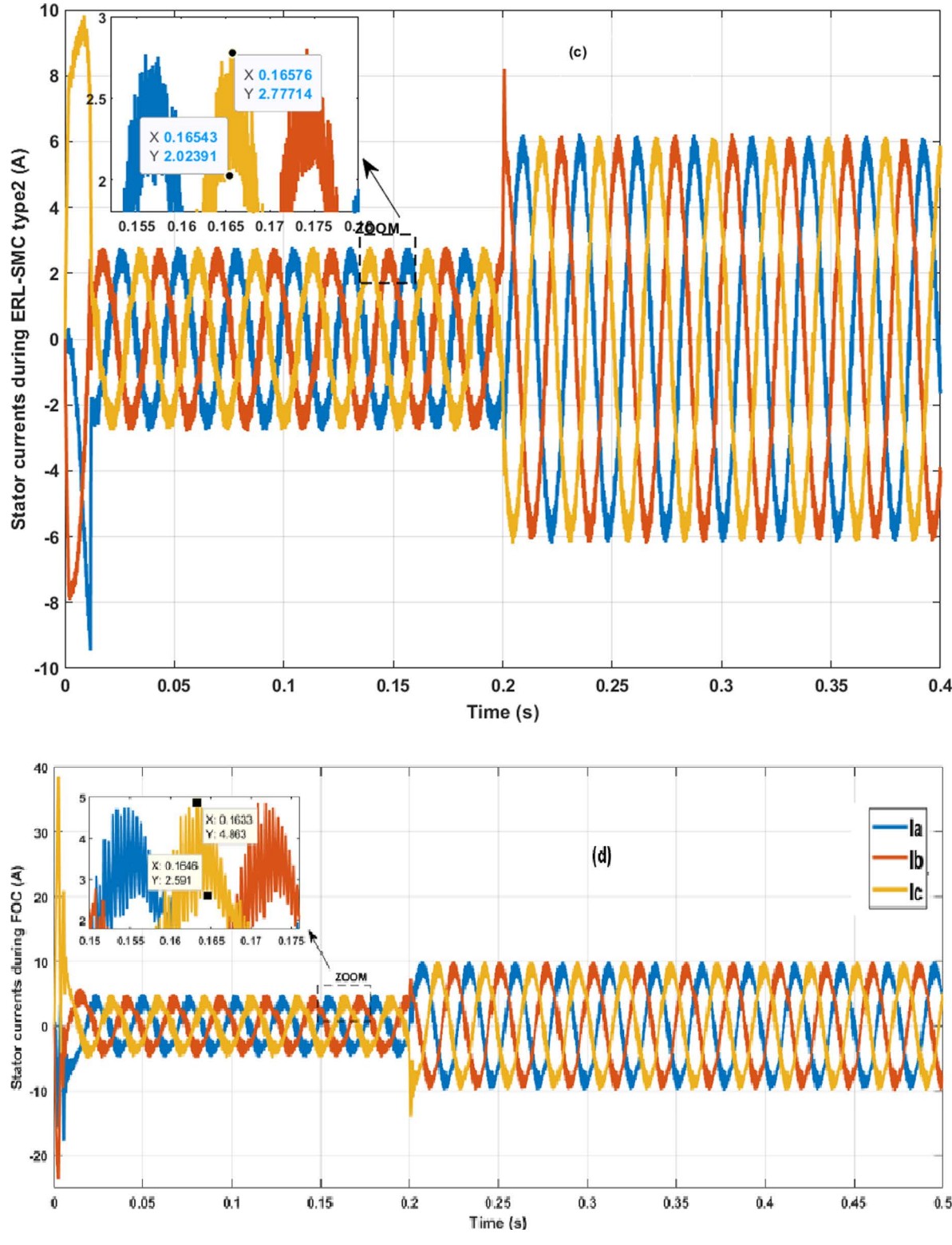

**Fig 20.** Continued.

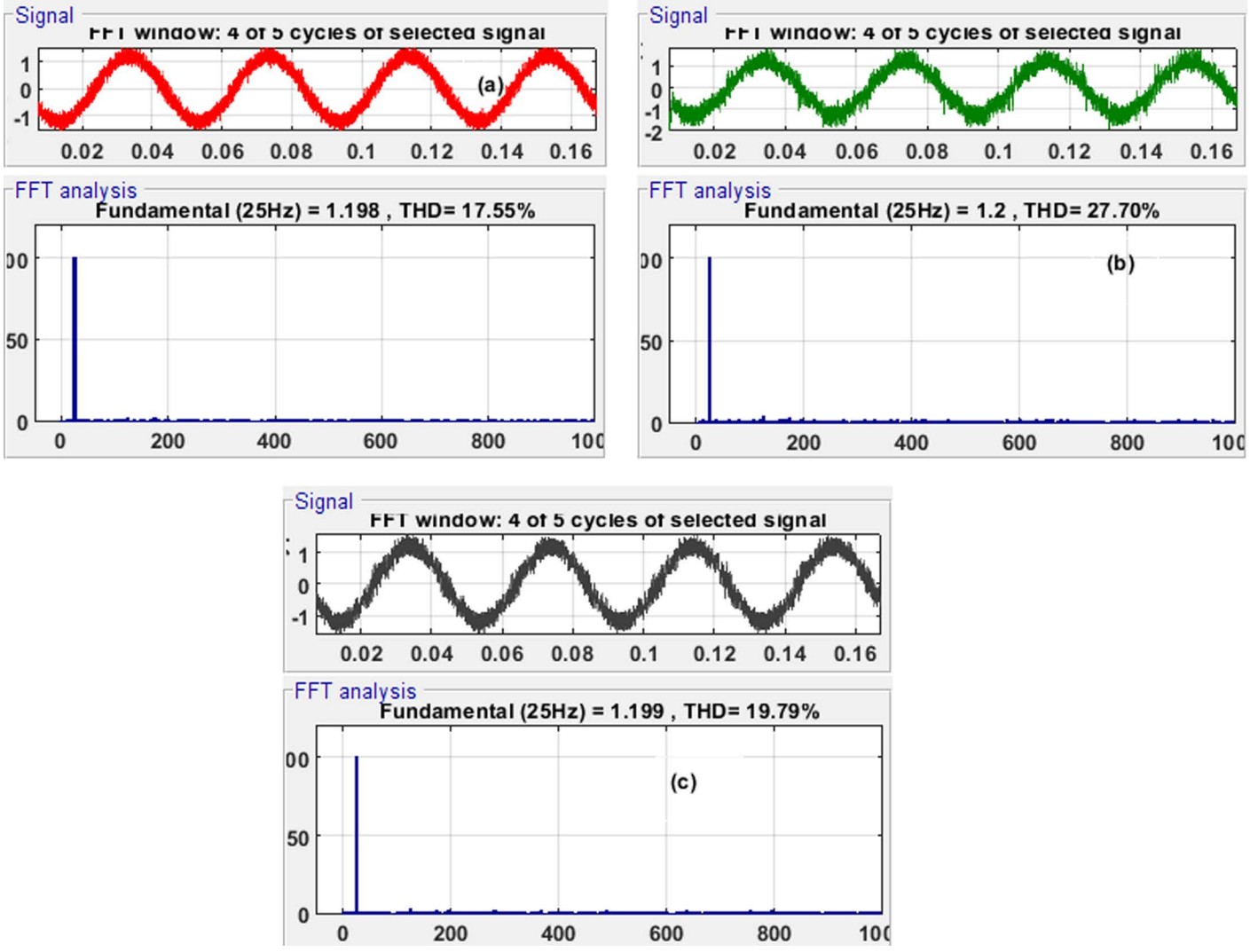

**Fig 21. Current Harmonic Analysis for MRAS with the three presented schemes.**

results validate the scheme's robustness in handling dynamic and challenging operating conditions, making it an excellent choice for sensorless speed control in high-performance motor applications. The comprehensive analysis solidifies the control-observation scheme (MRAS with ERL-SMC) as a highly effective and reliable solution for advanced PMSM control.

## 6. Conclusion

By addressing the challenges and proposing innovative sensorless control techniques, this work contributes to the advancement of PMSM control methods, fostering the development of more cost-effective, reliable, and efficient motor control systems. The investigation into the application of Model Reference Adaptive System (MRAS) to the observation of sensorless speed PMSMs combined with the nonlinear control design based on hybrid technique-based SMC has unveiled promising insights and potential advancements in the field of motor control. The primary objective of this work was to design and evaluate an observation technique system that allows a PMSM to emulate the behavior of a reference

model, accommodating uncertainties, variations, and disturbances. In essence, the novel design that combines the MRAS for PMSM sensorless and the SMC based on the ERL and CA holds great promise for addressing the challenges posed by uncertainties in motor dynamics. The achievements and lessons learned in this work contribute to the evolving landscape of advanced control strategies for PMSMs, opening avenues for more robust, reliable, and adaptive motor control systems in the future.

However, one notable disadvantage of the MRAS approach is its limited applicability in the low-speed region, where the performance and accuracy of speed estimation may degrade significantly. This limitation poses a challenge for applications requiring precise control at low speeds. Additionally, while the Pseudo-Sliding Mode Control with ERL reduces chattering, it may still introduce minor tracking errors under rapidly changing dynamic conditions. The complexity of implementing both MRAS and the hybrid SMC also increases the computational burden, which may limit the method's suitability for systems with strict real-time constraints or limited processing resources. Future work could focus on addressing these challenges, such as developing enhancements to improve MRAS performance at low speeds and optimizing the computational efficiency of the proposed hybrid control method. Experimental validations on real-world PMSM systems would also be beneficial, providing deeper insights and helping to refine the approach for practical and industry-ready applications.

## Author contributions

**Conceptualization:** Djaloul Karboua, Belgacem Toual, Toufik Mebkhouta, Youcef Chouiha, Zuhair A.Alqarni, Ahmad F. Tazay, Mohamed I. Mosaad.

**Data curation:** Djaloul Karboua, Said Benkaihoul.

**Formal analysis:** Djaloul Karboua, Said Benkaihoul, Youcef Chouiha, Zuhair A.Alqarni, Ahmad F. Tazay.

**Funding acquisition:** Djaloul Karboua, Zuhair A.Alqarni, Ahmad F. Tazay, Mohamed I. Mosaad.

**Investigation:** Djaloul Karboua, Said Benkaihoul, Zuhair A.Alqarni, Ahmad F. Tazay, Mohamed I. Mosaad.

**Methodology:** Djaloul Karboua, Belgacem Toual, Youcef Chouiha, Zuhair A.Alqarni, Ahmad F. Tazay, Mohamed I. Mosaad.

**Project administration:** Djaloul Karboua, Belgacem Toual, Toufik Mebkhouta, Mohamed I. Mosaad.

**Resources:** Djaloul Karboua, Youcef Chouiha, Zuhair A.Alqarni, Ahmad F. Tazay, Mohamed I. Mosaad.

**Software:** Djaloul Karboua, Said Benkaihoul, Toufik Mebkhouta.

**Supervision:** Belgacem Toual, Ahmad F. Tazay, Mohamed I. Mosaad.

**Validation:** Djaloul Karboua, Said Benkaihoul, Toufik Mebkhouta, Youcef Chouiha.

**Visualization:** Djaloul Karboua, Belgacem Toual, Said Benkaihoul, Youcef Chouiha, Zuhair A.Alqarni, Ahmad F. Tazay, Mohamed I. Mosaad.

**Writing – original draft:** Djaloul Karboua, Toufik Mebkhouta, Youcef Chouiha, Mohamed I. Mosaad.

**Writing – review & editing:** Djaloul Karboua, Toufik Mebkhouta, Youcef Chouiha, Zuhair A.Alqarni, Ahmad F. Tazay, Mohamed I. Mosaad.

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
