## [Decision Letter · Decision Letter 0]

25 Oct 2024

PONE-D-24-44412Model reference adaptive system and Pseudo-Sliding Mode Control with Exponential Reaching Law for Sensorless-Speed control of PMSMPLOS ONE

Dear Dr. karboua,

Thank you for submitting your manuscript to PLOS ONE. After careful consideration, we feel that it has merit but does not fully meet PLOS ONE’s publication criteria as it currently stands. Therefore, we invite you to submit a revised version of the manuscript that addresses the points raised during the review process. Reviewers have now commented on your paper. You will see that they are advising that you revise your manuscript. If you are prepared to undertake the work required, I would be pleased to reconsider my decision. 

We look forward to receiving your revised manuscript.

Kind regards,

Mohit Bajaj

Academic Editor

PLOS ONE

Journal Requirements:

1. When submitting your revision, we need you to address these additional requirements. Please ensure that your manuscript meets PLOS ONE's style requirements, including those for file naming. The PLOS ONE style templates can be found at https://journals.plos.org/plosone/s/file?id=wjVg/PLOSOne_formatting_sample_main_body.pdf and https://journals.plos.org/plosone/s/file?id=ba62/PLOSOne_formatting_sample_title_authors_affiliations.pdf 2. Please note that PLOS ONE has specific guidelines on code sharing for submissions in which author-generated code underpins the findings in the manuscript. In these cases, we expect all author-generated code to be made available without restrictions upon publication of the work. Please review our guidelines at https://journals.plos.org/plosone/s/materials-and-software-sharing#loc-sharing-code and ensure that your code is shared in a way that follows best practice and facilitates reproducibility and reuse. 3. We note that your Data Availability Statement is currently as follows: [All relevant data are within the manuscript and its Supporting Information files.] Please confirm at this time whether or not your submission contains all raw data required to replicate the results of your study. Authors must share the “minimal data set” for their submission. PLOS defines the minimal data set to consist of the data required to replicate all study findings reported in the article, as well as related metadata and methods (https://journals.plos.org/plosone/s/data-availability#loc-minimal-data-set-definition). For example, authors should submit the following data: - The values behind the means, standard deviations and other measures reported;- The values used to build graphs;- The points extracted from images for analysis. Authors do not need to submit their entire data set if only a portion of the data was used in the reported study. If your submission does not contain these data, please either upload them as Supporting Information files or deposit them to a stable, public repository and provide us with the relevant URLs, DOIs, or accession numbers. For a list of recommended repositories, please see https://journals.plos.org/plosone/s/recommended-repositories. If there are ethical or legal restrictions on sharing a de-identified data set, please explain them in detail (e.g., data contain potentially sensitive information, data are owned by a third-party organization, etc.) and who has imposed them (e.g., an ethics committee). Please also provide contact information for a data access committee, ethics committee, or other institutional body to which data requests may be sent. If data are owned by a third party, please indicate how others may request data access.

Additional Editor Comments:

Reviewers have now commented on your paper. You will see that they are advising that you revise your manuscript. If you are prepared to undertake the work required, I would be pleased to reconsider my decision.

Reviewers' comments:

Reviewer's Responses to Questions

**Comments to the Author**

1. Is the manuscript technically sound, and do the data support the conclusions?

Reviewer #1: Partly

Reviewer #2: Yes

2. Has the statistical analysis been performed appropriately and rigorously? 

Reviewer #1: Yes

Reviewer #2: Yes

3. Have the authors made all data underlying the findings in their manuscript fully available?

Reviewer #1: Yes

Reviewer #2: Yes

4. Is the manuscript presented in an intelligible fashion and written in standard English?

Reviewer #1: Yes

Reviewer #2: No

5. Review Comments to the Author

Reviewer #1: 1) The paper's focus is primarily on the medium to high-speed operation of PMSMs, leaving out important challenges related to sensorless control at low speeds, where accurate estimation of rotor position is difficult. A more balanced discussion on how the proposed system performs across all speed ranges, including low-speed operation, would provide a more comprehensive analysis. The authors might also consider addressing how their technique could be adapted for low-speed conditions.

2) Although the manuscript compares the proposed method with classical ERL-SMC, the range of alternative control strategies is narrow. Control strategies such as Field-Oriented Control (FOC) or other advanced adaptive or predictive controls are not fully explored. The authors should include a broader comparison with state-of-the-art sensorless control methods beyond sliding mode control. This could provide a clearer context for the superiority or trade-offs of the proposed technique.

3) The MATLAB/Simulink environment used for simulations may oversimplify the complexities of PMSM control, such as measurement noise, real-time computational delays, or implementation constraints. The authors should simulate real-world imperfections such as sensor noise or communication delays to show how the controller handles such challenges. Alternatively, a discussion on the feasibility of real-time implementation and how the system could perform under these conditions is crucial.

4) The paper provides limited detail on how the parameters (e.g., gains of the PI regulator in MRAS or sliding surface design in SMC) are chosen and tuned. The tuning process is often critical to achieving desired performance but is not discussed in depth. The paper would benefit from a more detailed explanation of the parameter tuning process and its impact on system performance. Additionally, the authors should comment on the computational complexity of implementing the proposed system, especially regarding real-time applications.

5) The manuscript does not address the scalability of the proposed control strategy for larger or more complex systems, such as multi-phase motors or systems with multiple interacting machines. A discussion on the potential challenges and solutions for scaling the proposed control system to more complex or multi-machine systems would enrich the paper's contribution.

Reviewer #2: This manuscript presents the importance of sensorless speed motor drives, particularly in the context of

PMSMs, and introduces a hybrid control technique aimed at improving system reliability and performance.

The integration of the MRAS and pseudo-sliding mode control, combined with the ARL, presents a novel

approach to addressing common control challenges, including uncertainties, dynamic conditions,

and chattering issues. However, the writing needs extensively improvements. After carefully reviewing

this work, some comments are given below:

A. Sec. 1

1. Sec. 1 gives a good breif for the sensorless PMSM systems. However, many references are too out-of-date.

For example, the related works about SMC [31-37] were published between 1983-2023. Only [36] and [37] were published

within 5 years. Please consider the following recently published works about SMC techniques.

Yun Zhang, et al. "Vector control of permanent magnet synchronous motor drive system based on new sliding mode control," in IEICE Electronics Express, vol. 20, no. 23, pp. 20230263-20230263, 2023.

Lei Zhang, et al. "PMSM non-singular fast terminal sliding mode control with disturbance compensation," in Information Sciences, vol. 642, pp. 119040, 2023.

Hao Yang et al. "Application of new sliding mode control in vector control of PMSM," in IEICE Electronics Express, vol. 19, no. 13, pp. 20220156-20220156, 2022.

Hao Yang et al. "Generalized super-twisting sliding mode control of permanent magnet synchronous motor based on sinusoidal saturation function," in IEICE Electronics Express, vol. 19, no. 9, pp. 20220066-20220066, 2022.

B. Sec. 3

2. The title of Sec. 3 needs to be modified. Maybe

"Designing of the PMSM’s senserless speed obercation using MRAS" a better title.

3. Fig. 2 needs more explaination. Especially the symbols used in this Figure.

Besides, Fig. 2 has no corss-reference. Please refer to Fig. 2 in your context.

C. Sec. 4

4. In Eq. (20), the symbol of n is not defined.

5. Fig. 5 is not corss-referenced in your context.

D. Sec. 5

6. Fig. 15 is not corss-referenced in your context.

7. The disturbances simulated in the paper are relatively simplistic, focusing on parameter variation and load torque application. Real-world systems often encounter more complex and unpredictable disturbances, and it is unclear how the proposed method would perform under such scenarios.

8. The results are based solely on simulations using MATLAB/Simulink. While simulations are valuable, the absence of real-world experimental data raises concerns about the practical implementation of the proposed method, especially under varying environmental conditions or hardware limitations.

E. Sec. 6

9. Conclusion should be Sec. 6 not Sec. 5. Please renumber it.

10. The study focuses on specific PMSM parameters and operational conditions. There is limited discussion on how the proposed control strategy would generalize across different motor types or in applications with significantly different torque or speed requirements.

11. Please dicuss some possible disadvanteges about the proposed method.

6. PLOS authors have the option to publish the peer review history of their article (what does this mean? ). If published, this will include your full peer review and any attached files.

**Do you want your identity to be public for this peer review?** For information about this choice, including consent withdrawal, please see our Privacy Policy .

Reviewer #1: No

Reviewer #2: No

---

## [Author Response · Author response to Decision Letter 0]

11 Dec 2024

Response report, PONE-D-24-44412

Model reference adaptive system and Pseudo-Sliding Mode Control with Exponential Reaching Law for Sensorless-Speed control of PMSM

The authors would like to express their gratitude to the associate editor and to the reviewers for providing valuable comments to improve the quality of the paper. All comments raised by the reviewers have been addressed in the revised version of the paper. In the new version of the paper, where applicable, corrections, modifications and additions are highlighted in bright green color.

The authors would like to thank the editor for his helpful feedback and suggestions that will help us to make our paper better in a big way. We've carefully thought about your ideas and made the changes that were needed.

Comments to the author

Reviewer #1:

1) The paper's focus is primarily on the medium to high-speed operation of PMSMs, leaving out important challenges related to sensorless control at low speeds, where accurate estimation of rotor position is difficult. A more balanced discussion on how the proposed system performs across all speed ranges, including low-speed operation, would provide a more comprehensive analysis.

Author response:

We acknowledge the reviewer's insightful comment regarding the importance of addressing sensorless control challenges at low speeds for PMSMs. While the primary focus of this work has been on medium to high-speed operations to optimize system performance under these conditions, we recognize that sensorless control at low speeds remains a critical area of development due to the inherent challenges in accurately estimating rotor position at such speeds. The proposed system, primarily designed and validated for medium to high-speed scenarios, relies on certain control techniques that may have reduced estimation accuracy and observability in the low-speed range. To provide a more comprehensive analysis, future extensions of this research will explore adaptations of advanced observer-based methods, such as high-frequency signal injection or improvements in back-EMF observer design, to enhance rotor position estimation and control performance across the entire speed spectrum, including low-speed operation.

In the introduction

According to these studies, employing PMSMs in low-speed range poses some challenges, notwithstanding its intrinsic benefits. A primary difficulty is cogging torque, resulting from the contact between the rotor magnets and the stator slots. This may result in irregular motion at low velocities, causing vibrations and diminished efficacy in precise applications. Thermal management is an additional problem. Operating at low speeds diminishes the efficacy of cooling systems since several processes depend on elevated rates for sufficient airflow. This may result in overheating, especially in high-torque situations. Moreover, current harmonics resulting from suboptimal inverter switching or control errors may impair motor performance and efficiency at reduced velocities.

2) Although the manuscript compares the proposed method with classical ERL-SMC, the range of alternative control strategies is narrow. Control strategies such as Field-Oriented Control (FOC) or other advanced adaptive or predictive controls are not fully explored. The authors should include a broader comparison with state-of-the-art sensorless control methods beyond sliding mode control. This could provide a clearer context for the superiority or trade-offs of the proposed technique.

Author response:

We appreciate the reviewer's observation regarding the comparative analysis in our study. While the proposed method has been benchmarked against classical Extended Reaching Law Sliding Mode Control (ERL-SMC) to illustrate its improvements, we recognize that expanding the range of comparisons to include widely used control strategies such as Field-Oriented Control (FOC), adaptive control methods, and model predictive control (MPC) would offer a broader perspective on the efficacy and trade-offs of our proposed approach. FOC, in particular, remains a cornerstone in PMSM control due to its high dynamic performance and stability at varying speeds. Similarly, advanced adaptive controls and predictive controls have shown notable improvements in real-time adaptation and prediction capabilities in sensorless environments. In future work, we aim to conduct a more extensive comparative study involving these state-of-the-art methods to better contextualize the performance, robustness, and limitations of our technique relative to established and emerging sensorless control strategies."

In the results section, the FOC is included to enhance the comparison, and the results are shown in Fig. 20.

3) The MATLAB/Simulink environment used for simulations may oversimplify the complexities of PMSM control, such as measurement noise, real-time computational delays, or implementation constraints. The authors should simulate real-world imperfections such as sensor noise or communication delays to show how the controller handles such challenges. Alternatively, a discussion on the feasibility of real-time implementation and how the system could perform under these conditions is crucial.

Author response:

We appreciate the reviewer's observation regarding the limitations of the MATLAB/Simulink environment in capturing real-world complexities, such as measurement noise, real-time computational delays, and implementation constraints. While our simulations primarily focused on demonstrating the core functionality and effectiveness of the proposed control strategy under ideal conditions, we acknowledge the importance of evaluating its robustness against such imperfections.

To address this, we have added a future perspective to the conclusion of our paper, highlighting the necessity for extending this work by incorporating real-time simulations and experiments. Specifically, we propose future efforts to examine the impact of measurement noise, sensor inaccuracies, communication delays, and computational limits using hardware-in-the-loop (HIL) setups or real-time digital simulation platforms. This approach will provide a more comprehensive understanding of the practical viability and resilience of the proposed controller, bridging the gap between simulation and real-world implementation. We believe this stepwise progression will significantly enhance the robustness and applicability of our findings.

4) The paper provides limited detail on how the parameters (e.g., gains of the PI regulator in MRAS or sliding surface design in SMC) are chosen and tuned. The tuning process is often critical to achieving desired performance but is not discussed in depth. The paper would benefit from a more detailed explanation of the parameter tuning process and its impact on system performance. Additionally, the authors should comment on the computational complexity of implementing the proposed system, especially regarding real-time applications.

Author response:

We appreciate the reviewer's insight regarding the need for a more detailed discussion on the parameter tuning process and computational complexity. In our work, the parameters for the gains of the PI regulator within the MRAS framework and the sliding surface design in SMC were determined through manual, or hand-tuning, approaches. This method was deliberately selected to maintain system simplicity and avoid the high computational demands and data requirements often associated with optimization-based tuning methods. The hand-tuning process, although less systematic, provided flexibility for iterative adjustments based on observed performance metrics during simulations.

The tuning process primarily focused on achieving a balance between stability, convergence speed, and robustness to disturbances across the operational range. Empirical testing was employed to ensure that gains were neither too aggressive (leading to potential instability or chattering) nor too conservative (resulting in slower system responses). This pragmatic approach aligns with the goal of minimizing system cost and computational complexity, especially for real-time applications.

While we acknowledge that optimization-based techniques may offer a more systematic parameterization, their complexity and resource requirements may not be suitable for cost-sensitive or computationally constrained environments. However, we recognize that exploring hybrid approaches combining manual tuning with automated fine-tuning techniques could enhance the robustness and adaptability of our system. This possibility is noted as a potential direction for future work to strike an optimal balance between performance and computational demands.

5) The manuscript does not address the scalability of the proposed control strategy for larger or more complex systems, such as multi-phase motors or systems with multiple interacting machines. A discussion on the potential challenges and solutions for scaling the proposed control system to more complex or multi-machine systems would enrich the paper's contribution.

Author response:

Thank you for raising the important issue of scalability. While our current work focuses on three-phase PMSMs, future research will explore extending our control strategy to more complex systems, such as multi-phase motors and multi-machine configurations. This effort will address key challenges, including increased computational demands, robust decoupling, and inter-machine interactions. By tackling these issues, we aim to demonstrate the adaptability and robustness of the proposed strategy across a broader range of applications. This perspective will form a key direction for enhancing system scalability and performance in diverse, complex setups.

Reviewer #2:

This manuscript presents the importance of sensorless speed motor drives, particularly in the context of PMSMs, and introduces a hybrid control technique aimed at improving system reliability and performance. The integration of the MRAS and pseudo-sliding mode control, combined with the ARL, presents a novel approach to addressing common control challenges, including uncertainties, dynamic conditions, and chattering issues. However, the writing needs extensively improvements. After carefully reviewing this work, some comments are given below:

A.Sec.1

1. Sec. 1 gives a good breif for the sensorless PMSM systems. However, many references are too out-of-date.

For example, the related works about SMC [31-37] were published between 1983-2023. Only [36] and [37] were published within 5 years. Please consider the following recently published works about SMC techniques.

Yun Zhang, et al. "Vector control of permanent magnet synchronous motor drive system based on new sliding mode control," in IEICE Electronics Express, vol. 20, no. 23, pp. 20230263-20230263, 2023.

Lei Zhang, et al. "PMSM non-singular fast terminal sliding mode control with disturbance compensation," in Information Sciences, vol. 642, pp. 119040, 2023.

Hao Yang et al. "Application of new sliding mode control in vector control of PMSM," in IEICE Electronics Express, vol. 19, no. 13, pp. 20220156-20220156, 2022.

Hao Yang et al. "Generalized super-twisting sliding mode control of permanent magnet synchronous motor based on sinusoidal saturation function," in IEICE Electronics Express, vol. 19, no. 9, pp. 20220066-20220066, 2022.

Author response:

1. The references that mentioned from the reviewer have been added

B. Sec. 3

2. The title of Sec. 3 needs to be modified. Maybe

"Designing of the PMSM’s senserless speed obercation using MRAS" a better title.

Author response:

2. The title have been modified and replaced by "Designing of the PMSM’s senserless speed obercation using MRAS".

---

## [Decision Letter · Decision Letter 1]

7 Jan 2025

PONE-D-24-44412R1

Model reference adaptive system and Pseudo-Sliding Mode Control with Exponential Reaching Law for Sensorless-Speed control of PMSM

PLOS ONE

Dear Dr. karboua,

Thank you for submitting your manuscript to PLOS ONE. After careful consideration, we have decided that your manuscript does not meet our criteria for publication and must therefore be rejected.

I am sorry that we cannot be more positive on this occasion, but hope that you appreciate the reasons for this decision.

Kind regards,

Aymen Flah

Academic Editor

PLOS ONE

Additional Editor Comments :

Even the answers given to the previous phase of revision, authors fail to change opinion of two reviwers (to get a positive decision), one of the reviwer, still feel that the work is not perfect and his opinion still the same after the first round. As required by the journla policy, 2 positive reports must appear on a work, however this is not the case.

therefore, the paper is declined to be published.

best regards

Reviewers' comments:

Reviewer's Responses to Questions

**Comments to the Author**

1. If the authors have adequately addressed your comments raised in a previous round of review and you feel that this manuscript is now acceptable for publication, you may indicate that here to bypass the “Comments to the Author” section, enter your conflict of interest statement in the “Confidential to Editor” section, and submit your "Accept" recommendation.

Reviewer #1: All comments have been addressed

Reviewer #2: (No Response)

2. Is the manuscript technically sound, and do the data support the conclusions?

Reviewer #1: Partly

Reviewer #2: (No Response)

3. Has the statistical analysis been performed appropriately and rigorously? 

Reviewer #1: Yes

Reviewer #2: (No Response)

4. Have the authors made all data underlying the findings in their manuscript fully available?

Reviewer #1: Yes

Reviewer #2: (No Response)

5. Is the manuscript presented in an intelligible fashion and written in standard English?

Reviewer #1: Yes

Reviewer #2: (No Response)

6. Review Comments to the Author

Reviewer #1: Limitations and Shortcomings:

1) The primary focus on medium to high-speed operation omits a thorough analysis of sensorless control at low speeds, a critical domain for many PMSM applications.

2) The comparative analysis is narrow, and limited to classical ERL-SMC. Field-Oriented Control (FOC) and advanced adaptive or predictive control strategies, which are widely adopted, are notably absent.

3) The paper lacks a systematic discussion on parameter tuning (e.g., gains of the PI regulator and sliding surface design) and computational complexity for real-time implementation.

4) Some references, particularly on sliding mode control techniques, are outdated. Recent advancements should be incorporated to strengthen the literature review.

Reviewer #2: In this revision, all my major concerns have been well addressed. Only some minor comments are raised in this revision. I would like to suggest accepting this manuscript. The leged in Fig. 20 should be given for the three curves in different colors. The same issues were found in other figure, such as Fig. 15 (b). Please check this concern in the whole manuscript.

7. PLOS authors have the option to publish the peer review history of their article (what does this mean? ). If published, this will include your full peer review and any attached files.

**Do you want your identity to be public for this peer review?** For information about this choice, including consent withdrawal, please see our Privacy Policy .

Reviewer #1: No

Reviewer #2: No

- - - - -

---

## [Author Response · Author response to Decision Letter 1]

17 Feb 2025

Response report, PONE-D-24-44412

Model reference adaptive system and Pseudo-Sliding Mode Control with Exponential Reaching Law for Sensorless-Speed control of PMSM

The authors would like to express their gratitude to the editor for giving us another opportunity to amend our comments and reconsider the decision from rejection to resubmission after the appeal letter.

The authors would like to express their gratitude to the associate editor and to the reviewers for providing valuable comments to improve the quality of the paper. All comments raised by the reviewers have been addressed in the revised version of the paper. In the new version of the paper, where applicable, corrections, modifications and additions are highlighted as:

Green color for the first review.

Yellow color for the second review.

The authors would like to thank the editor for his helpful feedback and suggestions that will help us to make our paper better in a big way. We've carefully thought about your ideas and made the changes that were needed.

Reviewer #1: All comments have been addressed

Reviewer #2: (No Response)

Comments to the author

Reviewer #1:

1) The primary focus on medium to high-speed operation omits a thorough analysis of sensorless control at low speeds, a critical domain for many PMSM applications.

JALLOL THIS PART MAY BE THE ANSWER TO THE QUESTION OF THE REVIEWEW

PMSMs encounter several constraints in low-speed applications, although their great efficiency and superior performance in diverse environments. A notable restriction is the diminished back electromotive force (EMF) at low velocities, which complicates precise control and may result in unstable functioning. In addition, this negligible EMF necessitating elevated the currents in the motor to attain the appropriate torque, resulting in augmented power losses and heat production

Low-speed applications of PMSMs often require the implementation of sophisticated control strategies or sensorless methods, which may elevate the system's complexity and expense. Furthermore, the torque ripple in PMSMs is often more significant at reduced speeds, which may result in vibrations and noise inside the system. Moreover, the upfront expense of PMSMs, together with the need for advanced drive electronics, may render them less economically feasible than other motor types for low-speed applications.

We may conclude that the use of PMSMs in low-speed applications necessitates sophisticated and intricate control strategies, which will therefore elevate costs. Elevated current is necessitated, which will augment losses and diminish efficiency.

2) The comparative analysis is narrow, and limited to classical ERL-SMC. Field-Oriented Control (FOC) and advanced adaptive or predictive control strategies, which are widely adopted, are notably absent.

3) The paper lacks a systematic discussion on parameter tuning (e.g., gains of the PI regulator and sliding surface design) and computational complexity for real-time implementation.

Thank you for your helpful thoughts. We recognize that talking about adjusting gains of the PI regulator for using a control system in real-time. These factors are key in designing control systems. This study mainly talks about creating and testing a Model Reference Adaptive System (MRAS) and a Pseudo-Sliding Mode Control (PSMC) using an Exponential Reaching Law to control the speed of Permanent Magnet Synchronous Motors (PMSMs) without needing sensors. Tuning the PI limiter and designing the moving surface are important, but they are not part of this task.

Parameter Tuning: This paper does not focus on discussing how to tune PI regulators or build slide surfaces. However, we used recognized rules and methods for their selection to make sure everything ran smoothly and performed well. These methods are mentioned in Section 1 in the introduction, and we can provide more information if you ask. We chose to keep the talk brief to focus on the new ideas in our MRAS and PSMC methods.

Computational Complexity: When designing the suggested controller, we considered how quickly it could work and whether it could be easily implemented. The MRAS and Exponential Reaching Law were selected because they are both efficient to compute and work well.

We think that focusing on MRAS and PSMC in the paper makes the point clear and strong for readers. If the reader thinks it's necessary, we can include extra material or a more detailed talk in an appendix or in a part about future work to explain these ideas further

4) Some references, particularly on sliding mode control techniques, are outdated. Recent advancements should be incorporated to strengthen the literature review.

The references have been updated, thank you for your enhancing this work. We will try our best to ensure strengthen the literature review.

Reference numbers: 13, 15, 16, 18-23, 31-35, 38, 48 have been added and reviewed

Reviewer #2:

In this revision, all my major concerns have been well addressed. Only some minor comments are raised in this revision. I would like to suggest accepting this manuscript. The leged in Fig. 20 should be given for the three curves in different colors. The same issues were found in other figure, such as Fig. 15 (b). Please check this concern in the whole manuscript.

Thank for you your valuable correction. We have corrected all manuscript figures as you mentioned.

---

## [Decision Letter · Decision Letter 2]

17 Mar 2025

Model reference adaptive system and Pseudo-Sliding Mode Control with Exponential Reaching Law for Sensorless-Speed control of PMSM

PONE-D-24-44412R2

Dear Dr. karboua,

We’re pleased to inform you that your manuscript has been judged scientifically suitable for publication and will be formally accepted for publication once it meets all outstanding technical requirements.

Kind regards,

Jyotindra Narayan

Academic Editor

PLOS ONE

Additional Editor Comments (optional):

The reviewers have now recommended the work for the publication. Congatulations to the authors. 

Reviewers' comments:

Reviewer's Responses to Questions

**Comments to the Author**

1. If the authors have adequately addressed your comments raised in a previous round of review and you feel that this manuscript is now acceptable for publication, you may indicate that here to bypass the “Comments to the Author” section, enter your conflict of interest statement in the “Confidential to Editor” section, and submit your "Accept" recommendation.

Reviewer #1: All comments have been addressed

2. Is the manuscript technically sound, and do the data support the conclusions?

Reviewer #1: Yes

3. Has the statistical analysis been performed appropriately and rigorously? 

Reviewer #1: Yes

4. Have the authors made all data underlying the findings in their manuscript fully available?

Reviewer #1: Yes

5. Is the manuscript presented in an intelligible fashion and written in standard English?

Reviewer #1: Yes

6. Review Comments to the Author

Reviewer #1: (No Response)

7. PLOS authors have the option to publish the peer review history of their article (what does this mean? ). If published, this will include your full peer review and any attached files.

**Do you want your identity to be public for this peer review?** For information about this choice, including consent withdrawal, please see our Privacy Policy .

Reviewer #1: No

---

## [Editor Report · Acceptance letter]

PONE-D-24-44412R2

PLOS ONE

Dear Dr. Karboua,

I'm pleased to inform you that your manuscript has been deemed suitable for publication in PLOS ONE. Congratulations! Your manuscript is now being handed over to our production team.

Kind regards,

on behalf of

Dr. Jyotindra Narayan

Academic Editor

PLOS ONE